# FROM TEXT TO TALK: AUDIO-LANGUAGE MODEL NEEDS NON-AUTOREGRESSIVE JOINT TRAINING

**Tianqiao Liu[1,2], Xueyi Li[1,\*], Hao Wang[3], Haoxuan Li[3], Zhichao Chen[3], Weiqi Luo[1], Zitao Liu[1]**
[1]Guangdong Institute of Smart Education, Jinan University
[2]TAL Education Group, [3] Peking University
`liutianqiao1@tal.com, lixueyi@stu2021.jnu.edu.cn`

## ABSTRACT

Recent advances in large language models have attracted significant interest in extending their capabilities to multimodal scenarios, particularly for speech-to-speech (S2S) conversational systems. However, existing multimodal models handling interleaved audio and text rely on autoregressive (AR) methods, overlooking that text depends on target-target relations whereas audio depends mainly on source-target relations. In this work, we propose Text-to-Talk (TtT), a unified audio-text framework that integrates AR text generation with non-autoregressive (NAR) audio diffusion in a single Transformer. By leveraging the any-order AR property of absorbing discrete diffusion, our approach provides a unified training objective for text and audio. To support this hybrid generation paradigm, we design a modality-aware attention mechanism that enforces causal decoding for text while allowing bidirectional modeling within audio spans, and further introduce three training strategies that reduce train-test discrepancies. During inference, TtT employs block-wise diffusion to synthesize audio in parallel while flexibly handling variable-length outputs. Comprehensive experiments on audio question answering, automatic speech recognition, automated audio caption and S2S benchmarks show that TtT consistently surpasses strong AR and NAR baselines, with additional ablation and training-strategy analyses confirming the contribution of each component. Our code and data are publicly available at `https://github.com/ai4ed/TtT`.

## 1 INTRODUCTION

The recent success of large language models (LLMs) has catalyzed a paradigm shift towards general-purpose multimodal large language models (MLLMs) capable of processing and generating information across diverse modalities (Xu et al., 2025; Team et al., 2023; Li et al., 2021). Among these, S2S conversational systems have emerged as a pivotal component in facilitating natural human-AI interaction. Conventional systems typically decompose this problem into a cascaded pipeline of automatic speech recognition (ASR), LLM-driven response generation, and text-to-speech (TTS) synthesis. While effective to a degree, this modular design introduces significant latency accumulation and error propagation between modules, hindering naturalness and real-world applicability. In response, recent end-to-end approaches like Moshi (Défossez et al., 2024), GLM-4-Voice (Zeng et al., 2024), and VITA-Audio (Long et al., 2025) have sought to unify speech understanding and generation within a single model. These models are typically trained through multi-stage pipelines that involve text-to-audio tokenizer training, interleaved data construction, text-audio alignment and task-oriented supervised fine-tuning (Huang et al., 2025; Li et al., 2025a; Ding et al., 2025; Chu et al., 2024; Kang et al., 2022). As shown in Figure 1, these methods aim to generate interleaved text and speech tokens in an autoregressive (AR) manner, which are then decoded into continuous audio waveforms by a separate neural codec or diffusion-based decoder (Mehta et al., 2025; Kong et al., 2020). However, this emerging paradigm faces a fundamental challenge. As illustrated in Figure 1, we identify a fundamental mismatch in prevailing approaches that employ a single language model to autoregressively generate both text and audio tokens while using identical AR training objectives across both modalities (Zeng et al., 2024; Xie & Wu, 2024; Borsos et al., 2023; Dang et al.,

---

*Corresponding author: Xueyi Li.

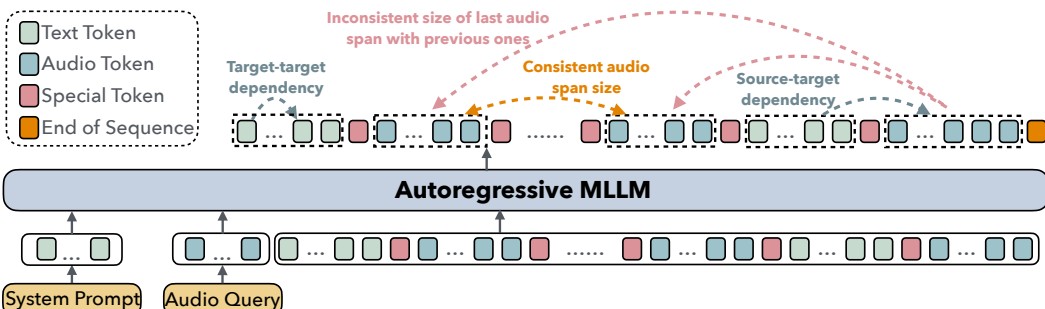

Figure 1: Distinct dependency structures for text and audio, and the resulting variable-length audio spans caused by disparate tokenization rates.

2024; Rubenstein et al., 2023). Such a uniform objective overlooks a critical distinction in their underlying generative processes. Text generation inherently follows a sequential causal structure characterized by strong **target-target** dependencies (Box et al., 2015), where each token explicitly conditions on previously generated tokens. Consequently, an incorrect token prediction can propagate and introduce subsequent errors due to the exposure bias inherent in AR models (Ranzato et al., 2015). In contrast, audio token generation is predominantly driven by **source-target** dependencies (Ren et al., 2020), where audio output primarily conditions on the source text rather than on the preceding audio tokens. Specifically, within the current non-autoregressive (NAR) span, audio token generation should remain faithful to the source text even when previous audio tokens are incorrectly predicted. Applying a purely AR objective to audio generation thus introduces unnecessary sequential constraints, leading to suboptimal training dynamics and magnifying error propagation.

This problem can be substantially alleviated by adopting a NAR generation strategy, which aligns better with the source-dependent nature of audio modeling. Recently, discrete diffusion has emerged as a compelling alternative to AR for discrete sequence modeling (Yu et al., 2025; Gong et al., 2025; Austin et al., 2021; Sahoo et al., 2024). Beyond empirical gains, recent theory shows that absorbing discrete diffusion can be interpreted as modeling the conditional distributions of clean tokens and admits a tight connection to any-order AR objectives (Ou et al., 2024). This raises a research question: *Can a unified model combining AR and NAR generation mitigate the mismatch introduced by applying a uniform AR objective to both text and audio?*

In this paper, we introduce Text-to-Talk (TtT), a unified audio-text MLLM that integrates AR text generation with NAR audio diffusion within a single Transformer initialized from a pretrained LLM. Text segments are trained with a standard AR cross-entropy objective, while audio segments are modeled via an NAR discrete diffusion process. During inference, the model dynamically switches between AR and NAR decoding strategies based on special control tokens. In summary, our work makes the following contributions:

- We identify and formalize the fundamental asymmetry in dependency structures between text and audio modalities. Leveraging the any-order AR nature of absorbing discrete diffusion, we establish a unified theoretical framework that proves our joint training objective provides an upper bound on the negative log-likelihood of the desired joint distribution.

- We propose a hybrid AR-NAR MLLM that seamlessly integrates AR text generation with discrete diffusion-based audio synthesis within a single Transformer initialized from a pretrained LLM. Our design preserves the reasoning and instruction-following capabilities of the base LLM while enabling efficient parallel audio generation.

- We introduce three principled training strategies to address the inherent train-test discrepancies in hybrid AR-NAR learning, enabling stable training and robust content-aware variable-length generation that bridges the gap between training and inference conditions.

- Extensive experiments across Audio-QA, ASR, AAC and S2S benchmark demonstrate that TtT consistently outperforms strong AR and NAR baselines, highlighting the advantage of the hybrid AR-NAR framework.

## 2 PRELIMINARY AND NOTATION

In this section, we establish the basic notation for interleaved audio-text sequences and provide brief overviews of the two core generative paradigms employed in our framework: AR modeling and absorbing discrete diffusion. These form the theoretical foundation of our proposed method in Section 3.

**Tokens, Vocabulary, and Interleaved Layout**   We consider interleaved discrete text–audio sequences of length $L$: $x = (x^1, \ldots, x^L)$ with a unified discrete vocabulary $\mathcal{V} = \mathcal{V}_{\text{text}} \cup \mathcal{V}_{\text{audio}} \cup \mathcal{S}$, where $x^l \in \mathcal{V}$ denotes the $l$-th token in the interleaved sequence $x$, $\mathcal{V}_{\text{text}}$ and $\mathcal{V}_{\text{audio}}$ are the discrete vocabularies for text tokens and audio tokens, respectively, and $\mathcal{S}$ contains special tokens such as $\langle \text{SOA} \rangle$ (start of audio), $\langle \text{EOA} \rangle$ (end of audio), $\langle \text{EOS} \rangle$ (end of sequence) and the absorbing mask token $[\mathbf{M}]$. A sequence $x$ is structured as a series of alternating text and audio spans: $x = (\mathcal{T}_1, \mathcal{A}_1, \ldots, \mathcal{T}_M, \mathcal{A}_M, \langle \text{EOS} \rangle)$, where $M$ denotes the number of text-audio span pairs, $m \in \{1, \ldots, M\}$ indexes the $m$-th pair, and the total length satisfies $L = \sum_{m=1}^{M} (|\mathcal{T}_m| + |\mathcal{A}_m|) + 1$, with the spans defined as follows:

- $\mathcal{T}_m = (t_{m,1}, \ldots, t_{m,|\mathcal{T}_m|}) \in (\mathcal{V}_{\text{text}} \cup \{\langle \text{SOA} \rangle\})^{|\mathcal{T}_m|}$ are text tokens.

- $\mathcal{A}_m = (a_{m,1}, \ldots, a_{m,|\mathcal{A}_m|}) \in (\mathcal{V}_{\text{audio}} \cup \{\langle \text{EOA} \rangle\})^{|\mathcal{A}_m|}$ are quantized audio tokens.

Let $f_\theta : \mathcal{V}^L \to \mathbb{R}^{L \times d}$ be a single Transformer. We use a shared output head $W \in \mathbb{R}^{d \times |\mathcal{V}|}$ (typically tied with input embeddings) to produce per-position logits over the entire vocabulary $\mathcal{V}$.

**AR Modeling**   AR models factorize the joint probability of a sequence $x = (x^1, \ldots, x^L)$ into a product of conditional probabilities, based on the chain rule: $p(x) = \prod_{i=1}^{L} p(x^i | x^{<i})$, where $x^{<i} = (x^1, \ldots, x^{i-1})$. This imposes a sequential, causal structure on the generation process. For a detailed discussion, please refer to Appendix A.3.1.

**Absorbing Discrete Diffusion**   Absorbing discrete diffusion models are a NAR paradigm for sequence generation. They consist of a forward process that corrupts a clean sequence by gradually replacing tokens with a special absorbing mask state $[\mathbf{M}]$, and a learned reverse process that aims to recover the original sequence from the corrupted input. A key insight from (Ou et al., 2024) is that the learning objective simplifies to modeling a time-independent conditional probability of the clean data. Specifically, the score for unmasking a token $v$ at a corrupted position is given by:

$$\underbrace{\frac{p_t(\ldots, \hat{x}^i = v, \ldots)}{p_t(\ldots, x^i = [\mathbf{M}], \ldots)}}_{\text{concrete score}} = \underbrace{\frac{e^{-\bar{\sigma}(t)}}{1 - e^{-\bar{\sigma}(t)}}}_{\text{time scalar}} \cdot \underbrace{p_0(v \mid x_{\text{vis}})}_{\text{clean conditional probability}} \tag{1}$$

where $i \in \{1, \ldots, L\}$ indexes the corrupted position with $x^i = [\mathbf{M}]$, and $\hat{x}$ denotes the same sequence as $x$ except that the $i$-th position is set to $v$ (i.e., $\hat{x}^i = v$). Here, $p_t(\cdot)$ denotes the marginal distribution of the forward corruption process at time $t$, $p_0(\cdot)$ is the clean data distribution, $x_{\text{vis}}$ denotes the visible (unmasked) tokens in the corrupted sequence (with their positions), and $\bar{\sigma}(t)$ is the cumulative corruption schedule. This denoising formulation is precisely the objective of an any-order AR model (AO-ARM), predicting a token given an arbitrary context of unmasked tokens. As demonstrated by (Ou et al., 2024), the diffusion training objective is mathematically equivalent to this AO-ARM objective, which averages the prediction loss over all possible permutations of the sequence: $\mathcal{L}_{AO}(\boldsymbol{x}_0) = \mathbb{E}_{\pi \sim U_\pi} \sum_{l=1}^{L} -\log q_\theta(x_0^{\pi(l)} \mid x_0^{\pi(<l)})$, where $\boldsymbol{x}_0 = (x_0^1, \ldots, x_0^L)$ is the clean token sequence, $\pi$ is drawn from $U_\pi$, the uniform distribution over all permutations of $\{1, \ldots, L\}$, $\pi(l)$ denotes the index at the $l$-th position of the permutation, $\pi(<l) = \{\pi(1), \ldots, \pi(l-1)\}$ denotes the set of indices revealed before step $l$, and $q_\theta(\cdot \mid \cdot)$ is the AO-ARM parameterized by $\theta$. Therefore, training an absorbing discrete diffusion model is equivalent to training a powerful ensemble of AR models that can operate in any order. More details are provided in Appendix A.3.2.

## 3 JOINT TEXT-AR & AUDIO-NAR MODEL

In this section, we introduce our proposed model, which integrates AR generation for text and discrete diffusion for audio within a single unified Transformer architecture.

## 3.1 AR Modeling for Text

We model text generation using a fixed, canonical AR order. Let $\pi_{\text{text}}$ denote the natural left-to-right permutation over all text token positions in the sequence, namely $\pi_{\text{text}}(1) < \pi_{\text{text}}(2) < \cdots < \pi_{\text{text}}(|\mathcal{T}_{\leq M}|)$, where $\mathcal{T}_{\leq M} = \cup_{m=1}^{M} \mathcal{T}_m$ collects all text token indices. At the span level, the probability of generating the $m$-th text span $\mathcal{T}_m = (t_{m,1}, \ldots, t_{m,|\mathcal{T}_m|})$ conditioned on all prior context is given by: $p_\theta(\mathcal{T}_m|\mathcal{T}_{<m}, \mathcal{A}_{<m}) = \prod_{j=1}^{|T_m|} p_\theta(t_{m,j}|\mathcal{T}_{<m}, \mathcal{A}_{<m}, t_{m,<j})$, where $t_{m,<j} = (t_{m,1}, \ldots, t_{m,j-1})$ is the prefix of text tokens within the current text span. To express the joint probability of all text tokens in the sequence, we account for the conditioning on preceding audio spans. The joint probability $p_\theta(x_{text})$ is therefore defined as the product of the probabilities of each text span, conditioned on all prior spans: $p_\theta(x_{text}) = \prod_{m=1}^{M} p_\theta(\mathcal{T}_m|\mathcal{T}_{<m}, \mathcal{A}_{<m}) = \prod_{m=1}^{M} \prod_{j=1}^{|\mathcal{T}_m|} p_\theta(t_{m,j}|\mathcal{T}_{<m}, \mathcal{A}_{<m}, t_{m,<j})$, the model is trained by minimizing the standard causal cross-entropy loss over all text positions:

$$\mathcal{L}_{\text{AR}}(x) = -\sum_{m=1}^{M} \sum_{j=1}^{|\mathcal{T}_m|} \log p_\theta\left(t_{m,j} \mid \mathcal{T}_{<m}, \mathcal{A}_{<m}, t_{m,<j}\right) \tag{2}$$

## 3.2 Absorbing Discrete Diffusion for Audio Spans

Building on the theoretical foundation established in Section 2, we apply absorbing discrete diffusion to audio spans $\mathcal{A}_{\leq M} = \cup_{m=1}^{M} \mathcal{A}_m$. This design choice aligns with the fundamental difference in dependency structures: audio tokens exhibit strong source-target dependencies (conditioning on source text), making them well-suited for the any-order AR nature of diffusion, while text tokens follow target-target causal dependencies, better handled by standard AR modeling.

**Audio-specific Corruption and Denoising**   For each training sample, we sample a masking level $\lambda \sim U([0, 1])$ and independently mask each audio token with probability $\lambda$, while preserving all text tokens. This creates corrupted sequences where audio spans contain a mixture of original tokens and mask tokens $[\mathbf{M}]$, but text spans remain intact. To enable efficient parallel training across all audio spans simultaneously, we apply masking operations to every audio span $\mathcal{A}_m$ in the sequence, rather than processing them sequentially. This parallel masking strategy significantly improves training efficiency while leveraging the time-independent nature of the denoising objective (Eq. 1).

**Training Objective for Audio Generation**   The model learns to predict the original audio tokens for masked positions by minimizing the $\lambda$-denoising cross-entropy loss over all audio spans. As discussed in (Ou et al., 2024), this objective is mathematically equivalent to the any-order AR objective, and can be equivalently expressed in the AO-ARM form:

$$\mathcal{L}_{\text{AO}}(x) = \sum_{m=1}^{M} \mathbb{E}_{\pi_m \sim U_{\pi_m}} \sum_{j=1}^{|\mathcal{A}_m|} -\log q_\theta(a_{m,\pi_m(j)}|\mathcal{T}_{\leq m}, \mathcal{A}_{<m}, a_{m,\pi_m(<j)}) \tag{3}$$

where $\pi_m$ is a random permutation over the positions within audio span $\mathcal{A}_m$, and $a_{m,\pi_m(<j)}$ denotes the audio tokens that appear before position $j$ in the permuted order. This formulation makes explicit that the audio generation objective is learning to predict each audio token conditioned on an arbitrary subset of other tokens within the same span, plus the full cross-modal context from text. This any-order AR nature is what enables parallel generation during inference.

## 3.3 Multimodal Factorization and Unified Objective

Having established AR modeling for text in Section 3.1 and discrete diffusion for audio in Section 3.2, we now formalize how these two paradigms can be unified within a single probabilistic framework. The key insight is to leverage the distinct dependency structures of each modality through a *partial-order factorization* that respects the causal nature of text while allowing flexible ordering within audio spans. Recall that text tokens exhibit strong target-target dependencies requiring causal ordering, while audio tokens primarily depend on source-target relationships with their corresponding text. This suggests that within each audio span $\mathcal{A}_m$, the tokens can be generated in any order as long as they condition on the appropriate cross-modal context $\mathcal{T}_{\leq m} \cup \mathcal{A}_{<m}$. We

formalize this intuition using partial orders over token positions. A partial order on a set $V$ is a binary relation $\preceq$ that is reflexive, antisymmetric, and transitive. A set equipped with such a relation is called a partially ordered set (poset). Two elements $a, b \in V$ are comparable if $a \preceq b$ or $b \preceq a$; otherwise, they are incomparable. An antichain is a subset of $V$ in which every pair of distinct elements is incomparable, so no internal ordering constraints exist among them (Davey & Priestley, 2002).

**Partial-order Formulation** Let $(V, \preceq)$ be a poset over all token indices in the sequence, where $V$ represents all token positions and $\preceq$ encodes precedence relationships. For our interleaved text-audio setting, we define: (1) Each text token $t_{m,j}$ precedes $t_{m,j+1}$ (maintaining left-to-right causality within text spans). (2) All tokens in span $m$ precede all tokens in span $m+1$ (maintaining cross-span dependencies). (3) Tokens within each audio span $\mathcal{A}_m$ form an antichain under $\preceq$ (no *mandatory* internal ordering), but the model is permitted to condition on previously generated tokens within the same span during training and inference under any linear extension. For any token $i$, let $\mathrm{Pa}(i)$ denote its set of predecessors under this partial order. By construction, each audio token $a_{m,j}$ has predecessors $\mathrm{Pa}(a_{m,j}) = \mathcal{T}_{\leq m} \cup \mathcal{A}_{<m}$, while for text tokens $\mathrm{Pa}(t_{m,j}) = \mathcal{T}_{<m} \cup \mathcal{A}_{<m} \cup t_{m,<j}$. Any linear extension $\ell$ of the partial order $(V, \preceq)$ induces a valid chain-rule factorization: $p(x) = \prod_{j=1}^{|V|} p(x_{\ell(j)} \mid x_{\mathrm{Pa}(\ell(j))})$. Since audio tokens within each span form an antichain, there are multiple valid linear extensions differing only in the within-span ordering of audio tokens. Rather than committing to a single extension, we can *marginalize* over all possible orderings within audio spans.

**Order-marginalized Factorization for Audio Spans** For an antichain $S \subseteq V$ (such as tokens within an audio span), we define the *order-marginalized conditional* by averaging over all permutations of $S$: $\tilde{p}_\theta(x_S \mid x_{V \setminus S}) = \mathbb{E}_{\pi \in \mathrm{Perm}(S)} \prod_{j \in S} q_\theta(x_{\pi(j)} \mid x_{V \setminus S}, x_{\pi(<j)})$, where $q_\theta(\cdot \mid \cdot)$ represents the any-order AR learned through discrete diffusion. When applied to our audio spans, this gives:

$$\tilde{p}_\theta(\mathcal{A}_m \mid \mathcal{T}_{\leq m}, \mathcal{A}_{<m}) = \mathbb{E}_{\pi_m \sim U_{\pi_m}} \prod_{j=1}^{|\mathcal{A}_m|} q_\theta(a_{m,\pi_m(j)} \mid \mathcal{T}_{\leq m}, \mathcal{A}_{<m}, a_{m,\pi_m(<j)}) \tag{4}$$

Intuitively, this averages the likelihood over all possible within-span orderings, reflecting the fact that audio tokens can be generated in any order given the appropriate cross-modal context. Note that while tokens within $\mathcal{A}_m$ form an antichain under the partial order (i.e., no mandatory sequential constraints), the order-marginalized conditional in Eq. 4 allows the model to leverage local target-target dependencies that may arise under specific generation orders. This flexibility enables the model to capture useful intra-span structures when beneficial.

**Hybrid AR-NAR Joint Distribution** Combining fixed-order AR for text with order-marginalized factorization for audio, our model induces the joint scoring function:

$$\tilde{p}_\theta(x) = \prod_{m=1}^{M} \left[ \underbrace{\prod_{j=1}^{|\mathcal{T}_m|} p_\theta(t_{m,j} \mid \mathcal{T}_{<m}, \mathcal{A}_{<m}, t_{m,<j})}_{\text{single-order AR for text}} \cdot \underbrace{\tilde{p}_\theta(\mathcal{A}_m \mid \mathcal{T}_{\leq m}, \mathcal{A}_{<m})}_{\text{order-marginalized any-order AR for audio}} \right] \tag{5}$$

This formulation reveals that both modalities are fundamentally AR: text uses a single linear extension (left-to-right), while audio integrates over all linear extensions consistent with the partial order.

**Training Objective and Upper Bound Analysis** In practice, we cannot directly optimize $\tilde{p}_\theta(x)$ because the order-marginalized conditional in Eq. 4 requires computing expectations over all permutations. Instead, we use the training objectives $\mathcal{L}_{\mathrm{AR}}(x)$ and $\mathcal{L}_{\mathrm{AO}}(x)$ derived in Section 3.2. The key theoretical insight is that our combined training objective provides a tight upper bound on the negative log-likelihood of the desired joint distribution. To see this, consider the audio term:

$$\mathbb{E}_{\pi_m \sim U_{\pi_m}} \sum_{j=1}^{|\mathcal{A}_m|} \left[ -\log q_\theta(a_{m,\pi_m(j)} \mid \mathcal{T}_{\leq m}, \mathcal{A}_{<m}, a_{m,\pi_m(<j)}) \right]$$

$$\geq -\log \mathbb{E}_{\pi_m \sim U_{\pi_m}} \prod_{j=1}^{|\mathcal{A}_m|} q_\theta(a_{m,\pi_m(j)} \mid \mathcal{T}_{\leq m}, \mathcal{A}_{<m}, a_{m,\pi_m(<j)}) \tag{6}$$

The right-hand side is precisely $-\log \tilde{p}_\theta(\mathcal{A}_m \mid \mathcal{T}_{\leq m}, \mathcal{A}_{<m})$ from Eq. 4. The left-hand side is exactly the audio loss term for span $m$ in our practical training objective $\mathcal{L}_{AO}(x)$.

To establish the unified upper bound, we now sum the inequality in Eq. 6 over all audio spans $m = 1, \ldots, M$:

$$\sum_{m=1}^{M} \mathbb{E}_{\pi_m} \sum_{j=1}^{|\mathcal{A}_m|} \left[ -\log q_\theta\big(a_{m,\pi_m(j)} \mid \mathcal{T}_{\leq m}, \mathcal{A}_{<m}, a_{m,\pi_m(<j)}\big) \right] \geq \sum_{m=1}^{M} \big( -\log \tilde{p}_\theta(\mathcal{A}_m \mid \mathcal{T}_{\leq m}, \mathcal{A}_{<m}) \big) \tag{7}$$

The left-hand side is exactly $\mathcal{L}_{AO}$. For the text terms, the $\mathcal{L}_{AR}$ loss is defined in Eq. 2. Combining the text and audio terms according to the joint factorization in Eq. 5 yields:

$$\mathcal{L}_{\text{Unified}}(x) \triangleq \mathcal{L}_{AR}(x) + \mathcal{L}_{AO}(x) \geq -\log \tilde{p}_\theta(x) \tag{8}$$

Detailed derivation of this inequality is provided in Appendix A.1.1, where this final inequality follows from combining the text equality with the audio inequality derived above. Thus, minimizing our practical training objective $\mathcal{L}_{\text{Unified}}(x)$ corresponds to minimizing an upper bound on the negative log-likelihood of the theoretically motivated joint distribution $\tilde{p}_\theta(x)$. This result is significant because: (1) It provides theoretical justification for our hybrid AR-NAR training approach. (2) It guarantees that optimizing the computationally tractable objective $\mathcal{L}_{\text{Unified}}(x)$ will not deviate arbitrarily from the theoretically optimal objective $-\log \tilde{p}_\theta(x)$.

**Training Pipeline and Loss Computation**  Our training pipeline starts from a pretrained text LLM and expands its vocabulary with discrete audio codebook tokens and control symbols ($\langle \text{SOA} \rangle$, $\langle \text{EOA} \rangle$). Each training sequence is organized as interleaved text spans and audio spans. Despite its theoretical and practical advantages, the hybrid AR-NAR paradigm introduces significant train-test discrepancies that can degrade generation quality. During training, audio spans are partially masked according to the diffusion process, while during inference, the model must generate audio and text tokens conditioned on complete text context and previously generated clean audio tokens. To bridge this gap, we propose three principled training strategies:

- **Batchwise AR & NAR Objective Mixing (BANOM)**: With probability $p_{\text{mix}}$, we skip the diffusion noise addition process for certain samples and compute gradients only on text tokens using AR loss. This ensures that, during training, text tokens occasionally observe clean, unmasked audio spans, which is consistent with the inference setting where text generation conditions on previously generated complete audio content rather than partially masked spans.

- **Prefix Preservation Masking (PPM)**: For a fraction $p_{\text{prefix}}$ of training samples, we randomly select a cutoff index $m$ and ensure that all preceding audio spans $\mathcal{A}_{<m} = \{\mathcal{A}_1, \ldots, \mathcal{A}_{m-1}\}$ remain unmasked, while applying NAR diffusion loss only to spans $\mathcal{A}_{\geq m} = \{\mathcal{A}_m, \mathcal{A}_{m+1}, \ldots, \mathcal{A}_M\}$. This strategy ensures that during training, when generating span $\mathcal{A}_m$, the model observes clean representations of all previous spans $\mathcal{A}_{<m}$, matching the inference scenario where audio spans are generated sequentially and each span $\mathcal{A}_m$ conditions on fully generated, clean preceding spans $\mathcal{A}_{<m}$ rather than their corrupted, partially masked versions.

- **Stochastic Span Truncation (SST)**: We address the positional bias in $\langle \text{EOA} \rangle$ prediction by randomly truncating audio span $\mathcal{A}_M$ during training. Due to disparate tokenization rates between text and audio, audio tokens significantly outnumber text tokens, resulting in fixed-size spans $\mathcal{A}_1, \ldots, \mathcal{A}_{M-1}$ and a variable-length final span $\mathcal{A}_M$. Since all audio spans undergo simultaneous diffusion training, the model learns to predict $\langle \text{EOA} \rangle$ at fixed positions for early spans, creating a strong positional bias that hinders content-aware termination learning for the final span. To mitigate this, we implement stochastic truncation: with probability $p_{\text{trunc}}$, we randomly select a truncation length $k < |\mathcal{A}_M|$ and create a truncated span $\mathcal{A}_M^{\text{trunc}} = (a_{M,1}, \ldots, a_{M,k})$ by removing the original $\langle \text{EOA} \rangle$ token and suffix tokens $(a_{M,k+1}, \ldots, a_{M,|\mathcal{A}_M|})$. This creates training samples where span termination occurs at arbitrary positions rather than fixed boundaries, forcing the model to predict $\langle \text{EOA} \rangle$ based on semantic content and contextual text rather than positional cues.

### 3.4 MODALITY-AWARE ATTENTION MECHANISM

Our attention design enforces a step-wise pattern across three content types: (1) The input prompt uses standard causal attention and serves as the shared condition for text and audio generation; (2)

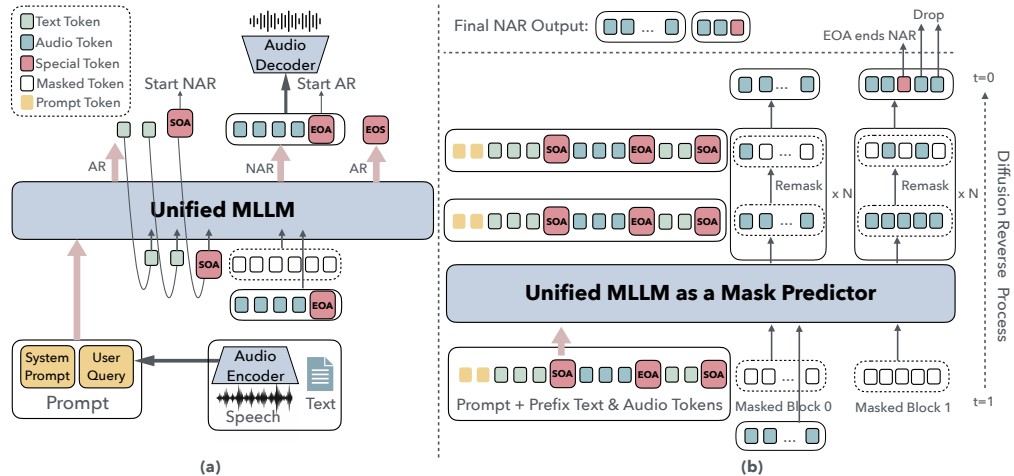

Figure 2: Overview of the proposed framework and its diffusion reverse process. (a) TtT framework. A unified MLLM that interleaves AR text and NAR audio generation. The model alternates between AR text decoding and NAR audio synthesis based on control tokens. (b) Diffusion reverse process. NAR audio generation through iterative denoising.

Text tokens $\mathcal{T}_m$ follow strict causal attention, attending to the prompt, all prior spans, and preceding tokens within the current text span, which is consistent with the AR text objective; (3) Audio tokens $\mathcal{A}_m$ use hybrid attention, attending causally to the prompt and earlier spans, while attending bidirectionally within the same audio span. This enables NAR diffusion to model each audio span as a whole, supports parallel audio-span training in a single forward pass, and prevents information leakage from future spans.

## 3.5 INFERENCE PROCESS

Figure 2 shows the overview of TtT and its inference process. During inference, TtT interleaves AR text decoding and NAR audio synthesis within a unified framework. Given an input audio, the model first follows the AR decoding process to generate text tokens. Once the control token ⟨SOA⟩ is produced, the model enters the NAR audio generation stage. In this stage, a masked audio block is initialized and iteratively refined through the diffusion reverse process, allowing multiple audio tokens within the same span to be generated in parallel. When ⟨EOA⟩ is predicted, the current audio span is considered complete, and any remaining tokens after ⟨EOA⟩ are discarded. The model then returns to AR text decoding and repeats this AR-NAR alternation until ⟨EOS⟩ is generated. Each completed audio span is immediately passed to the audio decoder, which enables low first-span latency and supports continuous streaming generation.

## 4 EXPERIMENTS

### 4.1 EXPERIMENTAL SETUPS

**Datasets** To effectively train and evaluate our proposed TtT framework, we follow prior work (Zeng et al., 2025; Ding et al., 2025; Long et al., 2025) and adopt a diverse collection of multi-task datasets, including ASR, TTS, audio chat, text chat, AAC, speech emotion classification (SEC), acoustic scene classification (ASC), and interleaved text-audio data. In total, the combined training corpus contains approximately 6.3 million samples across these tasks. For evaluation, we focus on three representative capabilities: (1) conversational reasoning via Audio-QA, (2) cross-modal alignment via ASR, and (3) audio comprehension via AAC. To further validate end-to-end S2S performance in realistic conversational scenarios, we additionally evaluate on URO-Bench (Yan et al., 2025), a comprehensive benchmark that integrates reasoning, understanding, and oral conversation tasks. Full details on training data composition and evaluation protocol are provided in Appendix A.4 and Appendix A.5.

Table 1: Comprehensive evaluation of TtT framework. Higher (↑) is better for Audio-QA, lower (↓) is better for ASR. Datasets abbreviations are available in Table 7.

| Models | Audio-QA (↑) | | | | ASR (↓) | | | | | |
|---|---|---|---|---|---|---|---|---|---|---|
| | AE. | LQ. | TQA. | WQ. | Fzh. | A2. | A1. | WS_m. | WS_n. | Fen. |
| *Main Results* | | | | | | | | | | |
| Qwen2.5-1.5B (AR) | 10.85 | 1.00 | 0.00 | 0.10 | 103.18 | 81.84 | 95.96 | 103.15 | 95.54 | 105.62 |
| Qwen2.5-1.5B (NAR) | 10.70 | 0.00 | 0.40 | 0.20 | 86.97 | 224.37 | 191.11 | 123.96 | 143.76 | 108.25 |
| TtT-1.5B (AR–NAR) | **15.68** | **23.75** | **3.47** | **7.70** | **44.36** | **14.89** | **16.72** | **52.23** | **41.52** | **49.00** |
| Qwen2.5-3B (AR) | 14.42 | 10.00 | 0.60 | 0.70 | 90.32 | 54.94 | 72.01 | 80.01 | 73.64 | 74.47 |
| Qwen2.5-3B (NAR) | 11.31 | 0.67 | 1.21 | 0.70 | 68.94 | 212.27 | 160.58 | 89.22 | 111.29 | 83.51 |
| TtT-3B (AR–NAR) | **17.46** | **34.68** | **6.53** | **11.61** | **55.67** | **12.53** | **13.65** | **53.83** | **44.29** | **64.31** |
| *Ablation Study* | | | | | | | | | | |
| TtT-3B w/o BANOM | 13.87 | 19.87 | 2.81 | 5.12 | 58.25 | 18.58 | 21.35 | 58.48 | 49.52 | 68.87 |
| TtT-3B w/o PPM | 14.27 | 22.79 | 2.71 | 5.54 | 58.86 | 15.63 | 18.83 | 57.76 | 47.92 | 67.37 |
| TtT-3B w/o SST | 14.12 | 10.20 | 1.30 | 3.72 | 56.39 | 25.43 | 31.03 | 64.41 | 56.70 | **62.60** |
| TtT-3B (AR–NAR) | **17.46** | **34.68** | **6.53** | **11.61** | **55.67** | **12.53** | **13.65** | **53.83** | **44.29** | 64.31 |
| *Training Strategy Comparison* | | | | | | | | | | |
| TtT-3B (AR–NAR) | 17.46 | 34.68 | 6.53 | 11.61 | 55.67 | 12.53 | 13.65 | 53.83 | 44.29 | 64.31 |
| Pretrain+AR | **29.45** | 15.93 | 3.61 | 11.45 | 23.37 | 9.79 | 12.67 | **26.75** | 20.91 | 19.49 |
| Pretrain+TtT | 26.73 | **40.07** | **11.07** | **21.43** | **18.99** | **6.80** | **5.78** | 27.59 | **19.85** | **19.10** |

**Evaluation**  To effectively evaluate our model on the above tasks, we carefully design the evaluation protocol and metrics: (1) For Audio-QA, we introduce an ASR-LLM pipeline that transcribes the model's spoken responses using language-specific ASR systems (Paraformer-zh for Chinese and Whisper-Large-v3 for English) and leverages a powerful LLM-as-a-Judge model (Qwen3-30B-A3B[1]) to assess semantic correctness against ground-truth answers; (2) For the ASR task, we directly measure transcription accuracy using word error rate (WER); (3) For the AAC task, we adopt the evaluation prompt from CLAIR-A (Wu et al., 2025), using thinking model Qwen3-30B-A3B to judge the caption quality; (4) For URO-Bench, we directly use the official evaluation code and protocol to ensure a fair comparison with existing systems. More details are in Appendix A.5.1.

**Baselines**  We compare TtT with state-of-the-art audio-language models, including Moshi Défossez et al. (2024), SpeechGPT Zhang et al. (2023), Kimi-Audio Ding et al. (2025), VITA-Audio Long et al. (2025), LLaMA-Omni Fang et al. (2025), GLM-4-Voice Zeng et al. (2024), Mini-Omni Xie & Wu (2024) and SLAM-Omni Chen et al. (2025a) (detailed descriptions in Appendix A.6).

**Model Configuration**  We adopt the Qwen2.5-Base model as the backbone, experimenting with parameter scales of 1.5B and 3B, and fine-tune all parameters during training. For the audio components, we directly follow the audio tokenizer and decoder design introduced in GLM-4-Voice (Zeng et al., 2024). These modules have been shown to provide efficient and high-quality speech tokenization and synthesis, and they allow our framework to leverage strong audio modeling without requiring additional architectural modifications. The training details are provided in Appendix A.7.

## 4.2 VALIDATING THE HYBRID AR-NAR ARCHITECTURE

To evaluate the effectiveness of our proposed TtT framework, we compare it with two representative variants, purely AR backbone and purely diffusion based NAR backbone. For fairness and scalability, all three frameworks are instantiated with backbones of 1.5B and 3B parameters.

**Performance Analysis on Audio-QA and ASR Tasks**  Table 1 (Main Results) provides the comparative results for the Audio-QA and ASR tasks. Our proposed TtT framework consistently outperforms both pure AR and NAR variants across all metrics. Specifically, at the 3B scale, TtT-3B surpasses Qwen2.5-3B (AR) by +3.04, +24.68, +5.93, and +10.91 on Audio-QA task. For ASR tasks, TtT-3B yields improvements of 42.41 and 58.36 absolute WER points over Qwen2.5-3B (AR). These substantial gains stem from our hybrid AR-NAR design: the NAR diffusion component enables efficient parallel denoising for tighter audio-text alignment, capturing audio's inherent

[1]https://huggingface.co/Qwen/Qwen3-30B-A3B-Thinking-2507

Table 2: Performance comparison on Audio-QA, ASR, and AAC tasks. Higher (↑) is better for Audio-QA and AAC; lower (↓) is better for ASR. Datasets abbreviations are available in Table 7.

| Models | Size | Audio-QA (↑) | | | | ASR (↓) | | | | | | AAC (↑) | |
|---|---|---|---|---|---|---|---|---|---|---|---|---|---|
| | | AE. | LQ. | TQA. | WQ. | Fzh. | A2. | A1. | WS_m. | WS_n. | Fen. | Clo. | MACS |
| *Large Models (> 7B)* | | | | | | | | | | | | | |
| Moshi | 7B | 25.63 | 48.30 | 16.75 | 16.85 | - | - | - | - | - | - | 4.32 | 12.01 |
| SpeechGPT | 7B | 10.00 | 30.96 | 16.53 | 24.53 | 101.45 | 120.77 | 111.81 | 123.15 | 124.86 | 45.15 | 2.10 | 3.95 |
| Kimi-Audio | 7B | 19.49 | **57.53** | 43.51 | 43.20 | **2.87** | **2.53** | **0.61** | **6.34** | **5.39** | **4.87** | **55.92** | **64.90** |
| VITA-Audio | 7B | 40.20 | 54.30 | 18.59 | 30.75 | 6.35 | 5.56 | 4.58 | 20.38 | 15.88 | 9.58 | 6.18 | 7.94 |
| LLaMA-Omni | 8B | 39.59 | 48.46 | 21.80 | 30.28 | - | - | - | - | - | - | 2.53 | 4.56 |
| GLM-4-Voice | 9B | **44.87** | **62.67** | **44.99** | **48.47** | - | - | - | - | - | - | 13.15 | 12.67 |
| *Efficient Models (≤ 3B)* | | | | | | | | | | | | | |
| Mini-Omni | 0.5B | 15.73 | 2.00 | 1.10 | 2.42 | 182.73 | 342.40 | 442.06 | 294.42 | 335.80 | 22.74 | 3.61 | 4.45 |
| SLAM-Omni | 0.5B | 17.47 | 24.75 | 3.51 | 7.90 | - | - | - | - | - | - | **54.52** | **50.46** |
| Qwen2.5-3B (AR) | 3B | 14.42 | 10.00 | 0.60 | 0.70 | 90.32 | 54.94 | 72.01 | 80.01 | 73.64 | 74.47 | 9.73 | 48.64 |
| Qwen2.5-3B (NAR) | 3B | 11.31 | 0.67 | 1.21 | 0.70 | 68.94 | 212.27 | 160.58 | 89.22 | 111.29 | 83.51 | 9.54 | 27.40 |
| TtT | 3B | 17.46 | 34.68 | 6.53 | 11.61 | 55.67 | 12.53 | 13.65 | 53.83 | 44.29 | 64.31 | 12.63 | 48.87 |
| Pretrain+TtT | 3B | **26.73** | **40.07** | **11.07** | **21.43** | 18.99 | **6.80** | **5.78** | **27.59** | **19.85** | **19.10** | 11.55 | 42.86 |

source-target dependencies, while AR text generation maintains coherent cross-modal conditioning and respects target-target dependencies. Purely NAR models perform notably worse due to order confusion from applying order-agnostic objectives to inherently sequential text-audio sequences.

**Ablation Study** To better understand the contribution of each training strategy in our hybrid AR-NAR framework, we perform an ablation study based on the full model TtT-3B (AR-NAR). The variant w/o BANOM corresponds to removing batchwise AR & NAR objective mixing from the full model, w/o PPM removes prefix preservation masking, and w/o SST removes stochastic span truncation. Table 1 (ablation study) presents the detailed results of our ablation experiments. From these results, we draw the following conclusions: All three training strategies have a positive impact on model performance, and removing any one of them leads to clear degradation on most metrics. For instance, on the LQ. dataset, removing SST reduces the score from 34.68 to 10.20. This drop occurs because stochastic truncation mitigates positional bias in ⟨EOA⟩ prediction, forcing span termination by semantic content rather than position. Removing it weakens variable-length audio generation and reduces flexibility in conversational outputs. Meanwhile, removing SST also substantially degrades ASR on major benchmarks, e.g., increasing WER on AISHELL-2 from 12.53 to 25.43 and on AISHELL-1 from 13.65 to 31.03.

**Effect of Multimodal Alignment Pretraining** To further investigate the effectiveness of our method on top of a multimodally aligned pretrained model, we perform large-scale multimodal pretraining based on the Qwen2.5-3B-Base model. Specifically, we construct a corpus of approximately 200B tokens covering ASR, TTS, text-only data, and interleaved text-audio data. The model is trained with a standard AR objective using a global batch size of 256 for 140k steps. This pretraining stage equips the backbone model (Qwen2.5-3B-Base) with strong cross-modal alignment ability before applying our hybrid AR-NAR learning framework. Table 1 (training strategy comparison) compares the AR-only and AR-NAR frameworks under two different training strategies, specifically training directly from Qwen2.5-3B-Base without multimodal pretraining (TtT-3B) and initialization from the multimodally aligned pretrained model (Pretrain+AR and Pretrain+TtT). From the table, we observe that: (1) When trained directly from Qwen2.5-3B-Base, our TtT framework achieves comparable or even superior performance to the AR-only baseline, indicating that the hybrid AR-NAR design is already competitive without pretraining; (2) when applied on top of the multimodally aligned pretrained model, Pretrain+TtT consistently matches or surpasses Pretrain+AR across both Audio-QA and ASR tasks. These results demonstrate that TtT not only performs strongly from scratch, but also benefits significantly when built upon large-scale multimodal alignment pretraining.

## 4.3 BENCHMARKING AGAINST STATE-OF-THE-ART MODELS

Having validated the effectiveness of our hybrid architecture and the benefits of multimodal pretraining, we now compare our best model (Pretrain+TtT) against state-of-the-art audio-language

Table 3: Evaluation results on URO-Bench. Higher (↑) is better for all tasks.

| Models | Size | Basic Task (↑) | | | Pro Task (↑) | | | Perceptual Quality (↑) | |
| | | Under-standing | Rea-soning | Oral Conversation | Under-standing | Rea-soning | Oral Conversation | NMOS | UTMOS |
|---|---|---|---|---|---|---|---|---|---|
| *Large Models (> 7B)* | | | | | | | | | |
| Moshi | 7B | 18.23 | 24.21 | 36.65 | 26.38 | 21.06 | 33.93 | 3.10 | 3.05 |
| SpeechGPT | 7B | 9.26 | 13.34 | 35.50 | 19.03 | 14.29 | 28.88 | 4.04 | 3.92 |
| Kimi-Audio | 7B | 83.89 | 53.88 | 54.44 | 53.25 | 41.44 | 50.17 | 3.52 | 2.93 |
| VITA-Audio | 7B | 52.08 | 51.45 | 54.97 | 32.36 | 54.77 | 45.81 | 3.95 | **4.24** |
| LLaMA-Omni | 8B | 53.71 | 41.93 | 64.05 | 34.66 | 51.51 | 43.91 | **4.09** | 4.00 |
| GLM-4-Voice | 9B | **85.82** | **61.63** | **69.90** | **55.47** | **51.89** | **61.30** | 3.86 | 4.15 |
| *Efficient Models (≤ 3B)* | | | | | | | | | |
| Mini-Omni | 0.5B | 15.01 | 14.80 | 29.71 | 23.51 | 33.09 | 33.46 | 4.15 | 4.42 |
| SLAM-Omni | 0.5B | 31.55 | 26.45 | 42.20 | 34.49 | 27.39 | 40.23 | **4.23** | **4.44** |
| Qwen2.5-3B (AR) | 3B | 34.32 | 13.15 | 23.68 | 16.32 | 34.99 | 25.90 | 3.96 | 4.16 |
| Qwen2.5-3B (NAR) | 3B | 7.22 | 10.12 | 20.01 | 12.59 | 13.70 | 25.64 | 3.47 | 2.35 |
| TtT | 3B | 43.39 | 24.00 | 30.08 | 23.37 | 33.78 | 34.82 | 3.89 | 4.25 |
| Pretrain+TtT | 3B | **57.63** | **39.30** | **45.68** | **32.38** | **43.76** | **46.10** | 3.90 | 4.23 |

models. Tables 2 and 3 group results by model scale, distinguishing efficient models from large ones. Notably, Moshi does not support ASR, and the official releases of LLaMA-Omni, SLAM-Omni and GLM-4-Voice lack ASR prompting, hence no ASR results are reported. Mini-Omni and SpeechGPT exhibit poor generalization to Chinese ASR tasks, as they are trained solely on English speech. Among efficient models, Pretrain+TtT achieves state-of-the-art performance across Audio-QA and ASR, while remaining competitive on AAC. It substantially outperforms 0.5B baselines such as Mini-Omni on Audio-QA and ASR, and SLAM-Omni on Audio-QA. While SLAM-Omni reports higher AAC scores (54.52 on Clotho, 50.46 on MACS), its official implementation relies on a separate 7B Vicuna model fine-tuned specifically for AAC. Notably, Pretrain+TtT also exceeds several 7B-scale models on some tasks. It outperforms SpeechGPT on all ASR benchmarks and on AE. and LQ. tasks in Audio-QA, and surpasses Moshi on WQ. and AE. tasks. These results demonstrate that our hybrid AR-NAR design enables a compact 3B model to match or exceed some significantly larger systems on several benchmarks.

Beyond the standard benchmarks, we further validate our model on URO-Bench, a comprehensive S2S benchmark that assesses speech understanding, reasoning, and oral conversation across basic and pro difficulty levels (Yan et al., 2025). As shown in Table 3, among efficient models, Pretrain+TtT achieves the best performance across both basic and pro difficulty levels. Compared to large models, Pretrain+TtT outperforms Moshi and SpeechGPT across all task categories, and achieves comparable performance to VITA-Audio and LLaMA-Omni on pro oral conversation tasks, while remaining competitive on pro reasoning tasks. While GLM-4-Voice achieves the strongest overall performance, the performance gap is reasonable given their ∼3× model size. The perceptual quality of both TtT and Pretrain+TtT falls within the 3.89-4.25 range (NMOS & UTMOS), confirming consistently good audio synthesis quality. However, Kimi-Audio exhibits notably lower perceptual quality despite its strong task completion performance. We attribute this degradation to language consistency issues observed in its generations. Kimi-Audio frequently produces mixed Chinese-English audio or generates Chinese audio for English tasks. While the semantic content may be correct, such cross-lingual inconsistencies significantly degrade perceptual audio quality.

## 5 CONCLUSION

In this work, we introduce a unified framework that combines AR text generation with NAR audio diffusion. By explicitly respecting the asymmetry between text and audio dependencies, our framework bridges the strengths of AR and NAR modeling within a single Transformer. We further propose simple yet effective strategies to mitigate train-test discrepancies, enabling robust and flexible audio generation. Experiments on Audio-QA, ASR, AAC and URO-Bench benchmarks demonstrate clear improvements over strong AR and NAR baselines. Our results highlight the importance of modality-aware design for building scalable and effective S2S systems.

ACKNOWLEDGEMENTS

This work was supported in part by Beijing Natural Science Foundation under Grant No. L252010; in part by NSFC under Grant No. 62477025; in part by Key Laboratory of Smart Education of Guangdong Higher Education Institutes, Jinan University (2022LSYS003), in part by the Excellent Graduate Student Cultivation Program of Jinan University under Grant No. 2024CXB040 and in part by the Guangdong University Student Science and Technology Innovation Cultivation Special Fund under Grant No. pdjh2026ak027.

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

# A    APPENDIX

## A.1    MATHEMATICAL DERIVATION

### A.1.1    DERIVATION OF THE TRAINING OBJECTIVE UPPER BOUND

Recall from Eq. 5 that the joint distribution factors as:

$$\tilde{p}_\theta(x) = \prod_{m=1}^{M} \left[ \prod_{j=1}^{|\mathcal{T}_m|} p_\theta\big(t_{m,j} \mid \mathcal{T}_{<m}, \mathcal{A}_{<m}, t_{m,<j}\big) \cdot \tilde{p}_\theta(\mathcal{A}_m \mid \mathcal{T}_{\leq m}, \mathcal{A}_{<m}) \right]. \quad (9)$$

Taking the negative logarithm of both sides gives:

$$-\log \tilde{p}_\theta(x) = -\sum_{m=1}^{M} \sum_{j=1}^{|\mathcal{T}_m|} \log p_\theta\big(t_{m,j} \mid \mathcal{T}_{<m}, \mathcal{A}_{<m}, t_{m,<j}\big) - \sum_{m=1}^{M} \log \tilde{p}_\theta(\mathcal{A}_m \mid \mathcal{T}_{\leq m}, \mathcal{A}_{<m})$$

$$= \mathcal{L}_{\text{AR}}(x) + \sum_{m=1}^{M} \left(-\log \tilde{p}_\theta(\mathcal{A}_m \mid \mathcal{T}_{\leq m}, \mathcal{A}_{<m})\right). \quad (10)$$

By Eq. 6 and its summation over $m$, we have:

$$\mathcal{L}_{\text{AO}}(x) \geq \sum_{m=1}^{M} \left(-\log \tilde{p}_\theta(\mathcal{A}_m \mid \mathcal{T}_{\leq m}, \mathcal{A}_{<m})\right). \quad (11)$$

Therefore, combining both components:

$$\mathcal{L}_{\text{AR}}(x) + \mathcal{L}_{\text{AO}}(x) \geq \mathcal{L}_{\text{AR}}(x) + \sum_{m=1}^{M} \left(-\log \tilde{p}_\theta(\mathcal{A}_m \mid \mathcal{T}_{\leq m}, \mathcal{A}_{<m})\right) = -\log \tilde{p}_\theta(x), \quad (12)$$

which establishes Eq. 8. This confirms that our practical training objective $\mathcal{L}_{\text{Unified}}(x)$ is a valid upper bound on the true negative log-likelihood, enabling tractable optimization while preserving consistency with the target joint distribution $\tilde{p}_\theta(x)$.

## A.2    RELATED WORK

### A.2.1    AUDIO-LANGUAGE MODEL PRETRAINING

Recent success of LLMs has been demonstrated through their exceptional performance in complex reasoning tasks, particularly in logical inference (Cheng et al., 2025; Liu et al., 2024) and mathematical problem-solving (Chen et al., 2025b; Liu et al., 2025b). Catalyzed by these advancements, there has been a paradigm shift towards general-purpose MLLMs capable of processing and generating information across diverse modalities (Zhu et al., 2025), along with broader considerations such as reliability and safety (Zhao et al., 2024; Li et al., 2025b; Zhao et al., 2025). Under this paradigm, recent advances in end-to-end audio-language models have moved beyond traditional cascaded architectures toward unified multimodal frameworks (Chen et al., 2022; Wang et al., 2023). Representative works include Moshi, which achieves real-time duplex speech conversation through hierarchical Transformer architectures (Défossez et al., 2024); GLM-4-Voice, which builds upon GLM-4-9B for robust Chinese and English speech processing (Zeng et al., 2024); and VITA-Audio, which introduces a lightweight multiple cross-modal token prediction (MCTP) module for fast audio-text generation with significantly reduced first-token latency (Long et al., 2025). More recent efforts have focused on scaling and production readiness: Step-Audio presents a 130B-parameter unified speech-text model with generative speech data engine and instruction-driven fine control across dialects, emotions, singing, and RAP (Huang et al., 2025), while Baichuan-Audio features text-guided aligned speech generation with multi-codebook discretization to preserve both semantic and acoustic information (Li et al., 2025a). UniWav proposes the first unified encoder-decoder framework that jointly learns representation encoders and generative audio decoders for both discriminative and generative speech tasks (Liu et al., 2025a).

A key limitation shared by these approaches is their reliance on uniform AR objectives for both text and audio tokens, which overlooks the distinct dependency structures of these modalities. Our work addresses this gap by proposing a hybrid AR-NAR framework that respects the inherent asymmetries between text and audio generation.

### A.2.2 DISCRETE DIFFUSION MODELS

Discrete diffusion models have emerged as a compelling alternative to AR generation, offering NAR approaches that can generate entire sequences in parallel (Chen et al., 2024). The foundational work of D3PMs generalized diffusion processes to discrete data through flexible transition matrices, with absorbing processes that progressively mask tokens proving particularly effective (Austin et al., 2021). This framework has since evolved through both theoretical advances and practical improvements. From a theoretical perspective, recent work has deepened our understanding of discrete diffusion dynamics. Ou et al. (2024) revealed that absorbing diffusion's concrete score can be expressed as time-independent conditional probabilities, leading to RADD, which is a reparameterized model that removes explicit time conditioning while establishing connections to any-order AR generation. Building on this foundation, Li & Cai (2025) formally characterized convergence rates, proving that KL divergence decays at $O(1/T)$ with bounds scaling linearly with token mutual information. However, Feng et al. (2025) identified a fundamental trade-off: while masked diffusion achieves near-optimal perplexity in constant steps, sequence-level tasks like reasoning may require steps linear in sequence length. Practical advances have focused on training efficiency and application domains. Shi et al. (2024) reformulated the variational objective as a weighted integral of cross-entropy losses, unifying prior approaches while achieving state-of-the-art results that even surpass comparable AR baselines. For complex reasoning tasks where AR models struggle with subgoal imbalance, Ye et al. (2025) demonstrated that multi-granularity diffusion modeling can achieve near-perfect accuracy by prioritizing harder subgoals during training. The scalability challenge has been addressed through innovative adaptation strategies. Rather than training from scratch, Gong et al. (2025); Nie et al. (2025) showed that pretrained AR models can be efficiently converted to diffusion models via continual pre-training, maintaining competitive performance while enabling parallel generation. Meanwhile, hybrid approaches are gaining traction: Lovelace et al. (2024) combined diffusion-based latent proposals with AR decoding for controllable generation, while Yang et al. (2025b) developed MMaDA, a unified multimodal diffusion foundation model that processes text, images, and reasoning within a single architecture.

### A.3 AR MODELING & ABSORBING DISCRETE DIFFUSION

#### A.3.1 AR MODELING

AR models are a fundamental class of generative models that factorize the joint probability distribution of a sequence $x = (x^1, \ldots, x^L)$ into a product of conditional probabilities, based on the chain rule:

$$p(x) = \prod_{i=1}^{L} p(x^i | x^{<i}) \tag{13}$$

where $x^{<i} = (x^1, \ldots, x^{i-1})$ represents the tokens preceding the current token $x^i$. This factorization imposes a sequential, causal structure on the generation process. Such models, typically implemented with Transformer decoders, are trained by minimizing the negative log-likelihood (NLL) of the data, which corresponds to a cross-entropy loss at each position.

#### A.3.2 ABSORBING DISCRETE DIFFUSION

Discrete diffusion models offer a NAR alternative for sequence generation. We focus on absorbing discrete diffusion (Austin et al., 2021; Ou et al., 2024), which involves a forward corruption process and a learned reverse denoising process.

**Forward Process** The forward process is a continuous-time discrete Markov chain that corrupts a clean sequence $x_0$ over a time interval $t \in [0, T]$. Its dynamics are governed by a time-dependent transition rate matrix $\boldsymbol{Q}_t = \sigma(t)\boldsymbol{Q}$, where $\sigma(t)$ is a positive noise schedule. For absorbing diffusion, the constant matrix $\boldsymbol{Q} = \boldsymbol{Q}^{\text{abs}}$ is defined as:

$$\boldsymbol{Q}^{\text{abs}}(x \to x') = \begin{cases} 1, & \text{if } x' = [\mathbf{M}] \text{ and } x \neq [\mathbf{M}], \\ -1, & \text{if } x' = x \neq [\mathbf{M}], \\ 0, & \text{otherwise.} \end{cases} \tag{14}$$

This structure dictates that any token $x \neq [\mathbf{M}]$ transitions to a special mask token $[\mathbf{M}]$ at a rate of $\sigma(t)$. The state $[\mathbf{M}]$ is an **absorbing state** because the transition rate out of it is zero (i.e., $\mathbf{Q}^{\text{abs}}([\mathbf{M}] \rightarrow x') = 0$ for all $x'$), meaning once a token is masked, it remains masked. Over time, the sequence converges to a fully masked state. The probability that a token is masked by time $t$ is given by $\lambda(t) = 1 - e^{-\int_0^t \sigma(s)ds}$.

**Reverse Process**  The reverse process is also a continuous-time Markov chain that learns to denoise a corrupted sequence $x_t$ back towards the clean data $x_0$. Its reverse transition rate matrix $\tilde{\mathbf{Q}}_t$ is related to the forward rate matrix by:

$$\tilde{\mathbf{Q}}_t(x_t \rightarrow \hat{x}_t) = \begin{cases} \mathbf{Q}_t(\hat{x}_t \rightarrow x_t)\frac{p_t(\hat{x}_t)}{p_t(x_t)}, & x_t \neq \hat{x}_t, \\ -\sum_{k \neq x} \tilde{\mathbf{Q}}_t(x_t, k), & \hat{x}_t = x_t. \end{cases} \tag{15}$$

The term $p_t(\hat{x}_t)/p_t(x_t)$ is known as the *concrete score*. Since the forward process only allows transitions to the $[\mathbf{M}]$ state, the only non-trivial reverse transitions are from $[\mathbf{M}]$ back to a vocabulary token. This simplifies the learning task to modeling the score for these specific denoising transitions.

**Time-Independent Score and the Denoising Objective**  A key theoretical insight for absorbing diffusion is that the concrete score analytically decomposes into a known, time-dependent scalar and a *time-independent* conditional probability over the clean data (Ou et al., 2024). Specifically, for a transition that unmasks position $i$ from $[\mathbf{M}]$ to a token $v$, the score is:

$$\underbrace{\frac{p_t(\ldots, \hat{x}^i=v, \ldots)}{p_t(\ldots, x^i = [\mathbf{M}], \ldots)}}_{\text{concrete score}} = \underbrace{\frac{e^{-\bar{\sigma}(t)}}{1 - e^{-\bar{\sigma}(t)}}}_{\text{time scalar}} \cdot \underbrace{p_0(v \mid x_{\text{vis}})}_{\text{clean conditional probability}}. \tag{16}$$

where $x_{\text{vis}}$ denotes the set of unmasked (visible) tokens in the corrupted sequence. This decomposition is crucial because it decouples the time-dependent dynamics from the data distribution. It implies that the model $q_\theta$ does not need to learn a complex function of time $t$. Instead, its sole objective is to learn to approximate the clean conditional distribution $p_0(v|x_{\text{vis}})$, which is a static, time-independent property of the data. The learning task is thus simplified to a denoising objective: given a corrupted sequence with some tokens masked, predict the original tokens for the masked positions based on the visible context.

**Equivalence to Any-Order AR Modeling**  This denoising perspective reveals a profound connection to AR modeling. A standard AR model learns to predict a token based on a fixed, causal context. The diffusion model, through its denoising objective, learns to predict a token given an arbitrary context of unmasked tokens. This ability to condition on any subset of the context is the defining feature of an any-order AR model (AO-ARM).

In fact, the principled training objective for the diffusion model, known as the $\lambda$-denoising cross-entropy loss, is mathematically equivalent to the training objective of an AO-ARM (Ou et al., 2024), which averages the prediction loss over all possible permutations (or orderings) of the sequence:

$$\mathcal{L}_{AO}(\boldsymbol{x}_0) = \mathbb{E}_{\pi \sim U_\pi} \sum_{l=1}^{d} -\log q_\theta(x_0^{\pi(l)}|x_0^{\pi(<l)}). \tag{17}$$

where $\pi$ is a random permutation of the token indices. Therefore, training an absorbing discrete diffusion model is equivalent to training a powerful ensemble of AR models that can operate in any order. This inherent flexibility is what enables parallel, NAR generation at inference time and makes it a suitable choice for modeling source-dependent modalities like audio.

## A.4  DATASET DETAILS

Table 4 provides a summary of the training datasets, with detailed examples provided in the Appendix A.7.1. During training, we aim to construct a balanced corpus that supports effective learning across multiple tasks. Specifically, we randomly sample one million instances from the ASR dataset, the TTS dataset, and the audio chat dataset respectively. In addition, we create bilingual interleaved

text and audio data, ensuring that Chinese and English are represented in approximately equal proportions. To build the audio chat corpus, we rely on the text-to-audio dataset VoiceAssistant-400K together with the text-based datasets OpenHermes-2.5 and Firefly-Train-1.1M, and we employ a TTS model, namely CosyVoice2, to convert text into synthetic audio so as to enrich the training data. To further enhance cross-modal alignment between text and audio, we follow prior work (Zeng et al., 2025) and supplement the training corpus with interleaved text and audio data derived from the large-scale pretrained corpus FineWeb-Edu. This strategy not only expands task coverage but also strengthens the model's ability to jointly learn from and align textual and acoustic modalities. The evaluation datasets are shown in Table 7.

Table 4: Summary of datasets used in training.

| Dataset | Language | Samples | Task Type |
|---|---|---|---|
| Emilia_zh | Chinese | 500000 | TTS |
| Emilia_en | English | 500000 | TTS |
| AISHELL2 | Chinese | | ASR |
| AISHELL3 | Chinese | | ASR |
| CommonVoice | Chinese, English | | ASR |
| GigaSpeech | English | 600000 | ASR |
| LibriSpeech | English | | ASR |
| MLS-Eng | English | | ASR |
| PeopleSpeech | English | | ASR |
| VoxPopuli | English | | ASR |
| WenetSpeech | Chinese | 400000 | ASR |
| VoiceAssistant-400K | English | | Audio Chat |
| OpenHermes-2.5 | English | 1000000 | Audio Chat |
| Firefly-Train-1.1M | Chinese | | Audio Chat |
| MathInstruct | English | 262039 | Text Chat |
| MACS | English | | AAC |
| Clotho-v2 | English | | AAC |
| Nonspeech7k | English | 59282 | SEC |
| VocalSound | English | | SEC |
| CochlScene | English | | ASC |
| Chinese-Fineweb-Edu (Skypile) | Chinese | 1500000 | Interleaved Data |
| FineWeb-Edu | English | 1500000 | Interleaved Data |
| **Total** | – | **6321321** | |

## A.5 EVALUATION DETAILS

### A.5.1 EVALUATION TASKS

**URO-Bench** We leverage URO-Bench for a more comprehensive evaluation of our proposed method against existing baselines (Yan et al., 2025). URO-Bench is specifically designed for audio-in, audio-out tasks, directly simulating real-world conversational scenarios. In this framework, the spoken outputs from each model are first transcribed into text using Whisper-large-v3 (Radford et al., 2023), and the resulting transcripts are then evaluated for correctness, coherence, and task alignment. This evaluation employs a hybrid scoring framework comprising three components: (1) an LLM-as-a-judge (originally implemented via commercial LLM APIs) to assess semantic correctness and task alignment; (2) rule-based metrics for automatic WER computation; and (3) a fine-tuned emotion-aware model to evaluate the appropriateness of affective expression in spoken responses. Together, these components ensure that model outputs are judged not merely on surface-level textual fidelity, but on semantic accuracy, transcription quality, and emotional coherence.

URO-Bench structures its evaluation across two difficulty levels: Basic and pro. Each level comprises three distinct categories: Understanding tasks, Reasoning tasks, and Oral Conversation tasks. This hierarchical design facilitates a fine-grained assessment of model capabilities across increasing levels of linguistic and cognitive demand, covering both single-round and multi-round scenarios. In our experiments, we report results on the English subset of URO-Bench's evaluation set. Due to

Table 5: WER performance of different ASR models on Chinese (zh) and English (en).

| Model | WER-zh ($\downarrow$) | WER-en ($\downarrow$) |
|---|---|---|
| Whisper-Large-v3 | 0.5054 | **0.2167** |
| Paraformer-zh | **0.1028** | 0.3946 |

limited access to the original evaluation APIs (Gemini Flash and GPT-4o-mini), we substitute the LLM-as-a-judge component with thinking model Qwen3-30B-A3B, while keeping all other components, including the Whisper-based ASR pipeline and rule-based scoring, identical to the original implementation.

Furthermore, the benchmark integrates a perceptual quality evaluation mechanism. We employ the strong UTMOS for the UTMOS score and DNSMOS for the NMOS score evaluation, enabling the joint assessment of content accuracy and acoustic quality (Saeki et al., 2022; Dubey et al., 2024). Importantly, none of the URO-Bench data was used during training or validation, ensuring an unbiased assessment of generalization.

**Audio-QA Task** In addition to URO-Bench, we also evaluate our model with the Audio-QA tasks established by Kimi-Audio (Ding et al., 2025). Previous evaluation framework in Kimi-Audio assesses Audio-QA performance using the text portion of interleaved outputs, which overlooks the fact that the audio output of an end-to-end speech model more directly reflects its ability to generate natural and semantically faithful responses. To address this limitation, we evaluate Audio-QA directly on the audio outputs of our framework by first applying an ASR model to transcribe the generated audio into text, where Whisper-large-v3 is used for English audio and Paraformer-zh for Chinese audio, with a comparison of ASR performance across different models provided in Table 5. The transcribed text is then combined with the original QA queries and the ground truth answers and passed to a large scale reasoning model, Qwen3-30B-A3B, which serves as an LLM-as-a-Judge model to determine whether the response semantically matches the reference and to provide either a correctness label or a graded score. We report the average accuracy or score on the benchmark, and this evaluation pipeline provides a more faithful assessment of our model's audio-to-audio QA ability in realistic conversational scenarios where speech serves as the output modality.

**ASR Task** To assess the model's capability in aligning speech with textual representations, we evaluate it on the ASR task, where the model generates text transcriptions from input audio and performance is measured using WER. A lower WER indicates more accurate recognition, which reflects not only strong ASR ability but also effective cross modal consistency achieved by our hybrid AR-NAR modeling framework.

**AAC Task** To assess the model's capacity to comprehend complex or acoustically challenging audios, we evaluate its audio captioning (AAC) performance on two established benchmarks: Clotho-v2 (Drossos et al., 2020) and MACS (Martín-Morató & Mesaros, 2021). We input audio clips and generate corresponding textual captions. The quality of these captions is then evaluated using Qwen3-30B-A3B (Yang et al., 2025a), guided by the evaluation prompt introduced in CLAIR-A (Wu et al., 2025), which emphasizes semantic relevance, completeness, and naturalness. The judge assigns a score on a 0-100 scale, where higher scores indicate better caption quality.

A.5.2 EVALUATION DATASETS

**URO-Bench** We use the English portion of URO-Bench Yan et al. (2025) to evaluate our model's performance. As detailed in Table 6, the benchmark consists of 10 basic tasks and 12 pro tasks. The basic tasks include 4 oral conversation, 4 reasoning, and 2 understanding tasks, while the pro tasks comprise 4 understanding, 4 reasoning, and 4 oral conversation tasks. The final score is obtained by first averaging the model's performance on each dataset, and then averaging these scores within each (difficulty, category) group.

**Audio-QA, ASR and AAC Tasks** We evaluate model performance on a diverse set of benchmarks covering both Audio Question Answering (Audio-QA), Automatic Speech Recognition (ASR), and automatic audio caption (AAC) tasks. For Audio-QA, we use four datasets: AlpacaEval, Trivi-

Table 6: Evaluation datasets used from URO-Bench.

| Dataset | Task /Evaluation Aspect | data nums | Category |
|---|---|---|---|
| *Basic tasks* | | | |
| AlpacaEval | Authentic, open-ended dialogue | 199 | Oral Conversation |
| CommonEval | Authentic, open-ended dialogue | 200 | Oral Conversation |
| WildchatEval | Real-world conversation | 349 | Oral Conversation |
| StoralEval | Deduce morals from a given story | 201 | Reasoning |
| Summary | Summarize a given story or statement | 118 | Oral Understanding |
| TruthfulEval | Factual questions about life | 470 | Reasoning |
| GaokaoEval | English listening questions | 303 | Understanding |
| Gsm8kEval | Practical mathematical problems | 582 | Reasoning |
| MLC | Mathematics, logic, and common sense | 177 | Reasoning |
| Repeat | Repeat the user's words verbatim | 252 | Understanding |
| *Pro Tasks* | | | |
| CodeSwitching-en | Understand code switching sentences | 70 | Understanding |
| GenEmotion-en | Respond in a specified tone | 54 | Oral Conversation |
| GenStyle-en | Respond in a specified style | 44 | Oral Conversation |
| MLCpro | Difficult mathematical, scientific questions | 91 | Reasoning |
| Safety-en | Reject answering privacy-related questions | 24 | Reasoning |
| SRT-en | Sing, recite poems, read tongue twisters | 43 | Oral Conversation |
| UnderEmotion-en | Understand the speaker's mood | 137 | Understanding |
| Multilingual | Respond in multiple languages | 1108 | Oral Conversation |
| ClothoEval-en | Comprehension of general ambient sounds | 265 | Understanding |
| MuChoEval-en | Comprehension of music | 311 | Understanding |
| MtBenchEval-en | Multi-round spoken dialogue | 190 | Reasoning |
| SpeakerAware-en | Multi-speaker multi-round dialogues | 55 | Reasoning |

aQA, and WebQuestions (English), along with LLaMAQuestions (English), assessing cross-lingual reasoning and comprehension from speech. For ASR, we include five datasets: Fleurs-zh/en (multilingual), AISHELL-1/2, and WenetSpeech (all Chinese), covering varied domains, accents, and recording conditions to robustly measure transcription accuracy. For AAC, we include two datasets: Clotho-v2 (English) and MACS (English), covering natural audios collected from various environmental sound clips, not limited to human-to-human dialogue, which helps assess the model's ability to understand the environment rather than simple language processing. Dataset details are summarized in Table 7.

Table 7: Evaluation datasets used for Audio-QA, ASR and AAC tasks.

| Dataset | Language | Task Type | Abbreviation |
|---|---|---|---|
| AlpacaEval | English | Audio-QA | AE. |
| LLaMAQuestions | English | Audio-QA | LQ. |
| TriviaQA | English | Audio-QA | TQA. |
| WebQuestions | English | Audio-QA | WQ. |
| Fleurs-zh | Chinese | ASR | Fzh. |
| AISHELL-2 | Chinese | ASR | A2. |
| AISHELL-1 | Chinese | ASR | A1. |
| WenetSpeech-test_meeting | Chinese | ASR | WS_m. |
| WenetSpeech-test_net | Chinese | ASR | WS_n. |
| Fleurs-en | English | ASR | Fen. |
| Clotho-v2 | English | AAC | Clo. |
| MACS | English | AAC | MACS |

## A.6 BASELINES

We compare our TtT model with the following state-of-the-art large audio-language models to evaluate its effectiveness:

- Moshi (Défossez et al., 2024): It unifies streaming speech and text understanding within a single AR framework, aligning acoustic and linguistic representations for low-latency real-time dialogue and robust multimodal instruction following.

- SpeechGPT (Zhang et al., 2023): It incorporates discrete speech tokens into a single language model and follows a three-stage training pipeline to enable unified speech-text understanding and cross-modal instruction following within one framework.

- Kimi-Audio (Ding et al., 2025): It uses a multi-task training pipeline to align speech, text and semantics through contrastive and generative objectives, enabling robust instruction-following and long-form audio dialogue understanding.

- VITA-Audio (Long et al., 2025): It tackles the latency bottleneck in LSLMs by introducing a fast interleaved decoding mechanism and dynamic token predictor, allowing efficient and streaming-capable audio response generation.

- LLaMA-Omni (Fang et al., 2025): It extends a unified language model to support real-time speech understanding and generation by integrating low-latency audio streaming, codec-based tokenization, and tightly aligned speech-text representations for seamless multimodal interaction.

- GLM-4-Voice (Zeng et al., 2024): It introduces a unified end-to-end spoken language model that interleaves speech and text modalities using a supervised speech tokenizer and joint training paradigm, enabling high-quality spoken dialogue generation.

- Mini-Omni (Xie & Wu, 2024): It enables real-time speech interaction by generating text and audio tokens in parallel within one model, using text-instructed parallel decoding and a lightweight training pipeline to preserve the base model's reasoning ability.

- SLAM-Omni (Chen et al., 2025a): It enables end-to-end spoken dialogue by modeling text and semantic audio tokens in parallel within a single model, supporting zero-shot timbre control and low-latency voice interaction through single-stage training.

## A.7 TRAINING DETAILS

We train our model using the AdamW optimizer with a global batch size of 2048, a learning rate of $2e^{-5}$, and a weight decay factor of $1e^{-2}$. The learning rate follows a cosine decay schedule with a linear warmup ratio of $0.01$. Training incorporates three stochastic strategies: (1) BANOM with probability 0.3; (2) PPM with ratio 0.3; (3) SST with probability 0.5. During inference, the model alternates between AR text decoding and NAR diffusion-based audio generation, where text decoding uses nucleus sampling with $k = 10$ and $p = 0.95$, and audio spans are generated with 200 diffusion steps, a block length of 32 tokens, and a total diffusion span length of 640 tokens under classifier-free guidance with scale 0.1. Since different training strategies lead to varying convergence speeds, reported results are based on checkpoints where training loss has converged. All experiments are conducted on 4 nodes with 8 NVIDIA A100 GPUs per node using the DeepSpeed runtime.

### A.7.1 DATA FORMAT OF TRAINING DATA

To enable unified training across diverse tasks, we transform all datasets into a consistent input-output format. On the one hand, this standardization allows the model to seamlessly integrate heterogeneous modalities such as speech, text, and interleaved audio-text sequences. On the other hand, a unified design is essential for supporting our training strategies, including batchwise AR & NAR objective mixing, prefix preservation masking, and stochastic span truncation. These strategies rely on a shared representation to operate across modalities in a consistent way. For clarity, we provide representative examples of the adopted data formats as follows, covering ASR, TTS, audio chat, text chat, AAC, SEC, ASC, and interleaved text-audio data, as shown in Figures 3 - 8.

### A.8 USAGE OF LLM

In this paper, the LLM is employed solely for text refinement, including correcting typos, fixing spelling errors, and enhancing readability. It is not used for generating research ideas, producing results, or creating original content.

---

### Automatic Speech Recognization (ASR) Data Format

```
"messages": [
        {
          "role": "system",
           "content": "You are an Automatic Speech Recognition (ASR) model. The user will
provide you with an audio input. Your task is to transcribe the audio into text and output the
result in an interleaved format: generate 13 text tokens followed by 26 audio tokens, and repeat
this pattern until the transcription is complete."
        },
        {
          "role": "user",
          "content": "<SOA>AUDIO_Sequence<EOA>"
        },
        {
          "role": "assistant",
          "content": "TEXT_Sequence_1<SOA>AUDIO_Sequence_1<EOA>
                      TEXT_Sequence_2<SOA>AUDIO_Sequence_2<EOA><EOS>"
        }
]
```

Figure 3: Example of ASR data format.

---

### Text-to-speech (TTS) Data Format

```
"messages": [
        {
        "role": "system",
        "content": "You are a Text to Speech (TTS) model. The user will provide you with a text
input. Your task is to transcribe the text into audio and output the result in an interleaved format:
generate 13 text tokens followed by 26 audio tokens, and repeat this pattern until the
transcription is complete."
        },
        {
          "role": "user",
          "content": "TEXT_Sequence"
        },
        {
          "role": "assistant",
          "content": "TEXT_Sequence_1<SOA>AUDIO_Sequence_1<EOA>
                      TEXT_Sequence_2<SOA>AUDIO_Sequence_2<EOA><EOS>"
        }
]
```

Figure 4: Example of TTS data format.

---

**Audio Chat Data Format**

"**messages**": [
   {
    "**role**": "system",
    "**content**": "You are an audio-to-audio chat model. The user will provide you with an audio input. Your task is to respond in an interleaved format: generate 13 text tokens followed by 26 audio tokens, and repeat this pattern until the response is complete."
   },
   {
    "**role**": "user",
    "**content**": "**<SOA>AUDIO_Sequence<EOA>**"
   },
   {
    "**role**": "assistant",
    "**content**": "**TEXT_Sequence_1<SOA>AUDIO_Sequence_1<EOA>**
      **TEXT_Sequence_2<SOA>AUDIO_Sequence_2<EOA><EOS>**"
   }
]

Figure 5: Example of audio chat data format.

---

**AAC/SEC/ASC Data Format**

"**messages**": [
   {
    "**role**": "system",
    "**content**": "You are a helpful audio model. The user will provide you with a text-based instruction and an audio input. Your task is to follow the instruction based on the audio and output the result in an interleaved format: generate 13 text tokens followed by 26 audio tokens, and repeat this pattern until the transcription is complete."
   },
   {
    "**role**": "user",
    "**content**": "**TEXT_Sequence<SOA>AUDIO_Sequence<EOA>**"
   },
   {
    "**role**": "assistant",
    "**content**": "**TEXT_Sequence_1<SOA>AUDIO_Sequence_1<EOA>**
      **TEXT_Sequence_2<SOA>AUDIO_Sequence_2<EOA><EOS>**"
   }
]

Figure 6: Example of AAC/SEC/ASC data format.

---

**Interleaved Data Format**

{
 "**text**": "**TEXT_Sequence_1<SOA>AUDIO_Sequence_1<EOA>**
   **TEXT_Sequence_2<SOA>AUDIO_Sequence_2<EOA><EOS>**"
}

Figure 7: Example of interleaved data format.

## Text Chat Data Format

```
"messages": [
        {
        "role": "system",
        "content": "You are a helpful model."
        },
        {
           "role": "user",
           "content": "TEXT_Sequence"
        },
        {
           "role": "assistant",
           "content": "TEXT_Sequence<EOS>"
        }
]
```

Figure 8: Example of text chat data format.

