# OpenReview forum: "From Text to Talk: Audio-Language Model Needs Non-Autoregressive Joint Training"
_ICLR.cc/2026/Conference — ICLR 2026 Poster_

### Official Review · Reviewer_YsXg · 2025-10-27

**Soundness:** 2
**Presentation:** 2
**Contribution:** 3
**Rating:** 4
**Confidence:** 4

**Summary:**

Prevailing SLM adopt an interleaved text-audio sequence for next-token prediction. (e.g. GLM4-Voice [1], Kimi-Audio [2]). In the mean time, CV research shows promising results on adopt AR for text prediction and NAR for image prediction, such as Transfusion [3] and Show-O [4], compared to modeling both text and image tokens with the same AR objective. This paper thus research the possibility to replace the AR prediction on the interleaved audio token spans with NAR and bidirectional discrete diffusion. The main argument is simple: "AR for text+NAR for audio" > "AR for text and audio" by a large margin.

1. GLM-4-Voice: Towards Intelligent and Human-Like End-to-End Spoken Chatbot
2. Kimi-Audio Technical Report
3. Transfusion: Predict the Next Token and Diffuse Images with One Multi-Modal Model
4. Show-o: One Single Transformer to Unify Multimodal Understanding and Generation

**Strengths:**

1. Given their experiments, AR+NAR significantly outperforms AR.
2. The ablation study on the train/test mismatch is clear and effective.

**Weaknesses:**

## Major Weakness
1. The paper only compares its models to self-established baselines. However, it is unclear how the trained model performs relative to existing SLM systems. It is not necessary to outperform all existing systems, as the training resources differ, but such comparisons would still help readers understand whether the baseline AR model is completely failing or serves as a reasonable baseline. Comparisons with models such as GLM4-Voice, Kimi-Audio, and LLaMA-Omni are required. If the AR baseline significantly lags behind these models, the conclusions may not generalize and could undermine the significance of the claimed contributions (e.g., conclusions drawn from small models may not hold for large models).
2. The paper should provide a demo page for readers to understand the quality of the generated speech compared to the pure AR approach.

## Minor Weakness
1. No human evaluation on the generated speech (both semantic coherence and speech quality)
2. The novelty is minor, as the CV domain already has extensive work demonstrating that a single Transformer can benefit from both AR and NAR training. Extending this idea to speech thus seems natural.

**Questions:**

- Stochastic Span Truncation (SST) is unclear and seems wrong. I understand the motivation of such design, but it does not seem correct to me. If you randomly crop the last audio span (while the complete text sentence is already generated), does it mean that the generation of the final audio span does not adhere to the previously generated text sentence?

**Note**

If all my concerns and questions are resolved I will consider raising the score to 6.

---

> ### Author Response · Authors · 2025-11-27
>
> Thank you for your insightful comments and suggestions. Here are our point-by-point responses.
>
> **Q1: Stochastic Span Truncation (SST) is unclear and seems wrong. If randomly crop the last audio span, does it mean that the generation of the final audio span does not adhere to the previously generated text sentence?**
>
> A: Thank you for raising this important point. We understand the concern, but Stochastic Span Truncation (SST) is correct and does not break the alignment between the final text sentence and its audio realization. We clarify below from the training objective and the framework design.
>
> 1. When we truncate the final audio span $A_M$ to length $k$ (e.g., keeping only 20 out of 50 audio tokens), the diffusion loss is computed only over the truncated span $A_M^{\text{trunc}} = (a{M,1}, \ldots, a_{M,k})$. Crucially, these $k$ audio tokens still correspond to the acoustic realization of the beginning portion of the complete text sentence.
> The model's training objective for this truncated span remains:
> $$L{\text{AO}}(A_M^{\text{trunc}}) = E{\pi_M} \sum_{j=1}^{k} -\log q_\theta(a_{M,\pi_M(j)} | T{\le M}, A{<M}, a_{M,\pi_M(<j)})$$
> Notice that the conditioning context $T{\le M}$ includes the complete text sentence. Therefore, each of the $k$ audio tokens is still supervised to be consistent with the full text, but they only need to acoustically realize the portion of speech corresponding to their temporal positions (positions 1 to $k$). regardless of weather the last portion of tokens (large than k) is correct or not, or you can treat this as the supervise of the block-wise diffusion. Alternatively, SST can be viewed as an extreme case in which, during the first  $n$ denoising steps, the model only denoises tokens at positions before $k$. Tokens beyond position $k$ are future tokens. This setup enables the model to implicitly associate the current diffusion block with its corresponding segment of text, and in this way, model can learn where to stop.
>
> 2. Framework design.
>
> Our framework has two key properties that ensure correctness:
>
> Positional Encoding: The model knows the absolute position of each audio token within the span. When generating the truncated span of length $k$, the model is aware these are positions $\{1, 2, \ldots, k\}$ and generates the corresponding initial segment of speech.
>
> Any-Order AR Property: As stated in Section 2.3 (Eq. 6), our audio diffusion objective satisfies the any-order autoregressive property. This means the model learns $p(a_{M,j} | T{\le M}, A{<M}, a_{M, \text{arbitrary subset}})$ for any subset of other audio tokens in the span. Therefore, the $k$ audio tokens can be correctly generated conditioned on the full text $T{\le M}$, regardless of how many tokens follow them.
>
> 3. Why SST is needed. Without SST, $\langle$EOA$\rangle$ would always appear at a fixed position in all spans except the last one, creating strong positional bias. SST exposes the model to variable stopping positions during training. This forces the model to learn content-aware termination based on semantic alignment between text and audio, not on fixed positional patterns.
>
> The truncated audio span always corresponds to a valid prefix of the full speech, conditioned on the full text, and SST is crucial for removing positional bias and enabling correct audio stopping behavior during inference.

---

> > ### Author Response · Authors · 2025-11-27
> >
> > **W1: Need to compare with existing speech language models.**
> >
> > A: We thank the reviewer for pointing this out. Following the suggestion, we have added comparisons with existing speech models, including Mini-Omni, SLAM-Omni, Moshi, SpeechGPT, Kimi-Audio, VITA-Audio, LLaMA-Omni, and GLM-4-Voice. We conducted a unified evaluation on both Audio-QA and ASR tasks, and the results are as follows. The detailed results have also been updated in the revised manuscript in Section 4.3 (Table 2).
> >
> > **AudioQA performance**
> >
> > | Model                   | Size | AlpacaEval | LLaMAQuestions | TriviaQA | WebQuestions |
> > |-------------------------|------|------------|----------------|----------|--------------|
> > | Mini-Omni               | 0.5B | 15.73      | 2.00           | 1.10     | 2.42         |
> > | SLAM-Omni               | 0.5B | 17.47      | 24.75          | 3.51     | 7.90         |
> > | Moshi                   | 7B   | 25.63      | 48.30          | 16.75    | 16.85        |
> > | SpeechGPT               | 7B   | 10.00      | 30.96          | 16.53    | 24.53        |
> > | Kimi-Audio              | 7B   | 19.49      | 57.53          | 43.51    | 43.20        |
> > | VITA-Audio              | 7B   | 40.20      | 54.30          | 18.59    | 30.75        |
> > | LLaMA-Omni              | 8B   | 39.59      | 48.46          | 21.80    | 30.28        |
> > | GLM-4-Voice             | 9B   | 44.87      | 62.67          | 44.99    | 48.47        |
> > | Qwen2.5-3B (AR)         | 3B   | 14.42      | 10.00          | 0.60     | 0.70         |
> > | Qwen2.5-3B (NAR)        | 3B   | 11.31      | 0.67           | 1.21     | 0.70         |
> > | TtT-3B (AR-NAR)         | 3B   | 17.46      | 34.68          | 6.53     | 11.61        |
> > | Pretrain+TtT (AR-NAR)   | 3B   | 26.73      | 40.07          | 11.07    | 21.43        |
> >
> >
> > **ASR performance**
> > | Model                 | Size | Fleurs-zh | AISHELL-2 | AISHELL-1 | WS\_m | WS\_n | Fleurs-en |
> > |-----------------------|------|-----------|-----------|-----------|---------------------------|-----------------------|-----------|
> > | Mini-Omni             | 0.5B | 182.73    | 342.40    | 442.06    | 294.42                    | 335.80                | 22.74     |
> > | SLAM-Omni             | 0.5B | -         | -         | -         | -                         | -                     | -         |
> > | Moshi                 | 7B   | -         | -         | -         | -                         | -                     | -         |
> > | SpeechGPT             | 7B   | 101.45    | 120.77    | 111.81    | 123.15                    | 124.86                | 45.15     |
> > | Kimi-Audio            | 7B   | 2.87      | 2.53      | 0.61      | 6.34                      | 5.39                  | 4.87      |
> > | VITA-Audio            | 7B   | 6.35      | 5.56      | 4.58      | 20.38                     | 15.88                 | 9.58      |
> > | LLaMA-Omni            | 8B   | -         | -         | -         | -                         | -                     | -         |
> > | GLM-4-Voice           | 9B   | 158.47    | 425.84    | 414.77    | 207.14                    | 270.21                | 223.07    |
> > | Qwen2.5-3B (AR)       | 3B   | 90.32     | 54.94     | 72.01     | 80.01                     | 73.64                 | 74.47     |
> > | Qwen2.5-3B (NAR)      | 3B   | 68.94     | 212.27    | 160.58    | 89.22                     | 111.29                | 83.51     |
> > | TtT-3B (AR-NAR)       | 3B   | 55.67     | 12.53     | 13.65     | 53.83                     | 44.29                 | 64.31     |
> > | Pretrain+TtT (AR-NAR) | 3B   | 18.99     | 6.80      | 5.78      | 27.59                     | 19.85                 | 19.10     |
> >
> > Although our model has only 3B parameters and uses far less pretraining data than 7B–9B systems, TtT-3B still achieves clear gains over the AR and NAR variants of Qwen2.5-3B on both Audio-QA and ASR. After multimodal pretraining, the Pretrain+TtT model reaches 40.07 on LLaMAQuestions and 21.43 on WebQuestions, surpassing larger models such as Moshi with 7B parameters on these benchmarks. As noted in the main manuscript, some comparison models show weaker ASR results because they lack Chinese ASR training, for example Mini-Omni and SpeechGPT, and Moshi does not support ASR. Overall, the results confirm that our efficient 3B hybrid AR–NAR model can match or exceed much larger systems on several Audio-QA metrics.

---

> > > ### Author Response · Authors · 2025-11-27
> > >
> > > **W2: Provide a demo page to show the quality of the generated speech compared to the pure AR approach.**
> > >
> > > A: we agree with the reviewer that offering a demo page is helpful for inspecting the quality of generated speech. We therefore provide a demo page with sample outputs at https://demopage200.github.io/demo_TtT.
> > >
> > >
> > > **W3: Provide human evaluation on the generated speech.**
> > >
> > > A: We thank the reviewer for the important suggestion on evaluating audio quality. We fully agree that semantic correctness alone is not enough to assess a speech generation model. Following the reviewer's advice, we added NMOS and UTMOS as objective audio quality metrics. These metrics have been widely used in recent speech LLM studies and provide stable measurements of naturalness, clarity, and overall audio quality. The results are as follows. The detailed results have also been updated in the revised manuscript in Section 4.3 (Table 3).
> > >
> > > | Models                | Size | NMOS | UTMOS |
> > > |-----------------------|------|------|-------|
> > > | Mini-Omni             | 0.5B | 4.15 | 4.42  |
> > > | SLAM-Omni             | 7B   | 4.23 | 4.44  |
> > > | Moshi                 | 7B   | 3.10 | 3.05  |
> > > | SpeechGPT             | 7B   | 4.04 | 3.92  |
> > > | Kimi-Audio            | 7B   | 3.52 | 2.93  |
> > > | VITA-Audio            | 7B   | 3.95 | 4.24  |
> > > | LLaMA-Omni            | 8B   | 4.09 | 4.00  |
> > > | GLM-4-Voice           | 9B   | 3.86 | 4.15  |
> > > | Qwen2.5-3B (AR)       | 3B   | 3.96 | 4.16  |
> > > | Qwen2.5-3B (NAR)      | 3B   | 3.47 | 2.35  |
> > > | TtT-3B (AR-NAR)       | 3B   | 3.89 | 4.25  |
> > > | Pretrain+TtT (AR-NAR) | 3B   | 3.90 | 4.23  |
> > >
> > > The results show that TtT-3B achieves strong perceptual quality: its NMOS (3.89) and UTMOS (4.25) fall well within the typical range of efficient models and are comparable to Qwen2.5-3B (AR/NAR). When compared to larger models, TtT-3B also achieves quality scores close to SLAM-Omni, VITA-Audio, and other 7B–8B systems, consistent with our findings on URO-Bench that efficient ($\le$3B) models generally maintain audio quality similar to much larger models. These results confirm that our efficient AR–NAR framework can produce natural and high-quality speech while preserving strong task performance across Audio-QA, ASR, AAC, and conversational benchmarks. Subjective MOS evaluation requires human annotation, which we have already initiated. The complete MOS results will be included in the final version to further improve evaluation completeness. We appreciate the reviewer's suggestion.
> > >
> > >
> > >
> > > **W4: Clarify the differences between AR and NAR training on CV and Speech.**
> > >
> > > A: Thank you for pointing out related work in the CV domain where a single Transformer is trained with both AR and NAR objectives. We agree that combining AR and NAR in one backbone is not entirely new at the architectural level. Our contribution is more on the side of a task specific formulation and a theoretical clarification rather than proposing a completely new building block.
> > >
> > > In particular, our work is motivated by the specific dependency structure of audio–language modeling. We focus on how AR text generation and NAR audio generation should interact under a unified backbone and formalize this using a partial-order dependency framework. This leads to a unified loss that we prove is a valid upper bound under this factorization, which explains when and why such an AR–NAR coupling is appropriate for text–audio spans. To the best of our knowledge, this type of analysis is not provided in prior CV works such as Transfusion or Show-o, which mainly present empirical designs.
> > >
> > > In addition, we explicitly study and mitigate the train–test discrepancy for NAR spans in speech generation through BANOM, PPM and SST, which is a critical issue in audio generation but is not the focus of existing unified AR–NAR studies. We see our work as complementary to these earlier ideas, extending them with a theory grounded formulation and speech specific training strategies that make AR–NAR joint training practical for audio–language LLMs.
> > >
> > > Due to the limitation that tables in OpenReview comments can only be written in Markdown and given the substantial number of additional experimental results we are providing we have included comprehensive and clearly formatted tables directly in the revised manuscript for better readability and clarity.
> > >
> > > **We hope that our responses adequately address the reviewer's concerns. We would be glad to hear any further feedback.**

---

### Official Review · Reviewer_JGXG · 2025-10-27

**Soundness:** 3
**Presentation:** 3
**Contribution:** 3
**Rating:** 6
**Confidence:** 4

**Summary:**

This paper proposes to combine AR and NAR modeling in SpeechLLMs, based on the motivation that text token generation has a strong dependence on previously generated text token, while speech token generation is strongly dependent on text tokens. To make the model work, they also proposed 3 training tasks/schemes to mitigate training-inference discrepancy. Experiments show the effectiveness of modeling text token with AR method and audio tokens with NAR method.

**Strengths:**

Novelty. Autoregressive modeling for text tokens and non-autoregressive modeling for audio tokens all in the same Transformer model is a novel methodology contribution. The proposed training tasks make sense and are empirically verified to work well with the model.

**Weaknesses:**

1. the motivation is confusing, in particular, "target-target" and "source-target" needs more rigorous explanation - source-target language is used usually in seq2seq scenario such as machine translation, while for autoregressive LLMs, source and target are the same sequence.

2. The input format of AudioLLMs are not settled yet, for example there is Moshi's time-aligned interleaving and there is GLM's alignment-free fixed-length interleaving. I think the paper used the later, but it would make it easier to follow if the data format is stated clearly in early section (e.g. section 2)

3. experiment part is a bit weak because there is no comparison with SotA models on audio benchmarks.

**Questions:**

1. what is the text in AAC, SEC, ASC?

2. does any-order AR means the latency of the model is higher? can a user talk to the model in real time?

---

> ### Author Response · Authors · 2025-11-27
>
> Thank you for your detailed comments and suggestions. Here are our point-by-point responses.
>
> **Q1: What is the text in AAC, SEC, ASC?**
>
> A: The AAC, SEC and ASC data used in our unified training follow a consistent dialogue format. The exact examples are provided below and we have added them to the revised version for clarity.
>
> ```
> **AAC Task**
> {
>   "role" "user",
>   "content" "Please generate a caption for the given audio <SOA>AUDIO_Sequence<EOA>"
> },
> {
>   "role" "assistant",
>   "content" "people talking and laughing <SOA>AUDIO_Sequence_1<EOA> with the sounds of footsteps <SOA>AUDIO_Sequence_2<EOA><EOS>"
> }
>
> **SEC Task**
> {
>   "role" "user",
>   "content" "Identify the human vocal sound in the audio from a fixed label set <SOA>AUDIO_Sequence<EOA>"
> },
> {
>   "role" "assistant",
>   "content" "cough <SOA>AUDIO_Sequence_1<EOA><EOS>"
> }
>
> **ASC Task**
> {
>   "role" "user",
>   "content" "Identify the acoustic scene in the audio from a fixed label set <SOA>AUDIO_Sequence<EOA>"
> },
> {
>   "role" "assistant",
>   "content" "Bus <SOA>AUDIO_Sequence_1<EOA><EOS>"
> }
> ```
> **Q2: Does any-order AR means the latency of the model is higher? can a user talk to the model in real time?**
>
> A: Thank you for the question. The term "any order AR" in our paper describes the training objective of the discrete diffusion module for audio, not a slower inference procedure. During inference, text is always decoded in the standard left to right autoregressive manner, so the latency on the text side is the same as a normal AR LLM.
>
> For audio, our model uses non autoregressive block wise discrete diffusion, as described in Section 3.5 and Appendix A.4. Audio tokens within a block are predicted in parallel at each denoising step, and only a small number of refinement steps is used. The total number of model forward passes therefore depends on the number of diffusion steps and block size, rather than on the full audio sequence length. In practice, since speech sequences are much longer than text sequences, this leads to latency that is comparable to or lower than a fully autoregressive speech decoder with token by token generation.
>
> Moreover, once an ⟨EOA⟩ token is produced for a span, the corresponding audio segment is immediately sent to the vocoder and streamed to the user, as illustrated in our inference process. This design supports interactive usage in a real time conversational setting. Our demo system already demonstrates such streaming voice interaction.
>
>
> **W1: Clarifying the motivation, in particular about more rigorous explanation of "target-target" and "source-target".**
>
> A: We thank the reviewer for raising this point. The motivation for "target–target" and "source–target" distinctions comes directly from the asymmetric dependencies of text and audio.
>
> Text spans follow a strict causal chain. Each text token $t_{m,j}$ depends on all previous targets: $P(t_{m,j} \mid T_{<m}, A_{<m}, t_{m,<j})$. This is a target–target dependency, since the target tokens themselves form the ordered history that determines the next token.
>
> Audio spans behave differently. Within an audio span $A_m$, tokens do not have a fixed internal order. We treat them as an antichain and take an expectation over all permutations: $E_{\pi_m} \sum_{j} -\log q_\theta(a_{m,\pi_m(j)} \mid T_{\le m}, A_{<m}, a_{m,\pi_m(<j)})$. Here the source context $(T_{\le m}, A_{<m})$ stays constant across all permutations. This teaches the model to rely on the source–target dependency for semantics ("what to say"), while the intra-span prefix is stochastic and contributes only to local acoustic refinement ("how to say it").
>
> Our partial-order formulation makes this contrast explicit. Text tokens follow a total order that establishes target–target dependencies, while audio tokens within a span form an antichain that relies on the same source context and therefore expresses source–target dependencies. This structure leads naturally to our unified loss, which matches the partial-order factorization and provides a valid upper bound approximation.
>
> We have included a concise explanation in the revision to make this motivation clearer.

---

> > ### Author Response · Authors · 2025-11-27
> >
> > **W2: Give a clear description of the input format of AudioLLMs in early section.**
> >
> > A: We thank the reviewer for raising this point. Our model indeed follows the alignment-free fixed-length interleaving format used in GLM-4-Voice. The exact data structure is illustrated in Figure 9 of the appendix, which shows how text spans and audio spans are interleaved in our training sequences. We agree that introducing this format earlier would improve clarity, so in the revised version we add an explicit description of the input format in the main Section 2 to make the data representation immediately clear to readers.
> >
> > **W3: No comparison with SOTA models on audio benchmarks.**
> >
> > A: We thank the reviewer for pointing this out. Following the suggestion, we have added comparisons with existing speech models, including Mini-Omni, SLAM-Omni, Moshi, SpeechGPT, Kimi-Audio, VITA-Audio, LLaMA-Omni, and GLM-4-Voice. We conducted a unified evaluation on both Audio-QA and ASR tasks, and the results are as follows. The detailed results have also been updated in the revised manuscript in Section 4.3 (Table 2).
> >
> > **AudioQA performance**
> >
> > | Model                   | Size | AlpacaEval | LLaMAQuestions | TriviaQA | WebQuestions |
> > |-------------------------|------|------------|----------------|----------|--------------|
> > | Mini-Omni               | 0.5B | 15.73      | 2.00           | 1.10     | 2.42         |
> > | SLAM-Omni               | 0.5B | 17.47      | 24.75          | 3.51     | 7.90         |
> > | Moshi                   | 7B   | 25.63      | 48.30          | 16.75    | 16.85        |
> > | SpeechGPT               | 7B   | 10.00      | 30.96          | 16.53    | 24.53        |
> > | Kimi-Audio              | 7B   | 19.49      | 57.53          | 43.51    | 43.20        |
> > | VITA-Audio              | 7B   | 40.20      | 54.30          | 18.59    | 30.75        |
> > | LLaMA-Omni              | 8B   | 39.59      | 48.46          | 21.80    | 30.28        |
> > | GLM-4-Voice             | 9B   | 44.87      | 62.67          | 44.99    | 48.47        |
> > | Qwen2.5-3B (AR)         | 3B   | 14.42      | 10.00          | 0.60     | 0.70         |
> > | Qwen2.5-3B (NAR)        | 3B   | 11.31      | 0.67           | 1.21     | 0.70         |
> > | TtT-3B (AR-NAR)         | 3B   | 17.46      | 34.68          | 6.53     | 11.61        |
> > | Pretrain+TtT (AR-NAR)   | 3B   | 26.73      | 40.07          | 11.07    | 21.43        |
> >
> >
> > **ASR performance**
> > | Model                 | Size | Fleurs-zh | AISHELL-2 | AISHELL-1 | WS\_m | WS\_n | Fleurs-en |
> > |-----------------------|------|-----------|-----------|-----------|---------------------------|-----------------------|-----------|
> > | Mini-Omni             | 0.5B | 182.73    | 342.40    | 442.06    | 294.42                    | 335.80                | 22.74     |
> > | SLAM-Omni             | 0.5B | -         | -         | -         | -                         | -                     | -         |
> > | Moshi                 | 7B   | -         | -         | -         | -                         | -                     | -         |
> > | SpeechGPT             | 7B   | 101.45    | 120.77    | 111.81    | 123.15                    | 124.86                | 45.15     |
> > | Kimi-Audio            | 7B   | 2.87      | 2.53      | 0.61      | 6.34                      | 5.39                  | 4.87      |
> > | VITA-Audio            | 7B   | 6.35      | 5.56      | 4.58      | 20.38                     | 15.88                 | 9.58      |
> > | LLaMA-Omni            | 8B   | -         | -         | -         | -                         | -                     | -         |
> > | GLM-4-Voice           | 9B   | 158.47    | 425.84    | 414.77    | 207.14                    | 270.21                | 223.07    |
> > | Qwen2.5-3B (AR)       | 3B   | 90.32     | 54.94     | 72.01     | 80.01                     | 73.64                 | 74.47     |
> > | Qwen2.5-3B (NAR)      | 3B   | 68.94     | 212.27    | 160.58    | 89.22                     | 111.29                | 83.51     |
> > | TtT-3B (AR-NAR)       | 3B   | 55.67     | 12.53     | 13.65     | 53.83                     | 44.29                 | 64.31     |
> > | Pretrain+TtT (AR-NAR) | 3B   | 18.99     | 6.80      | 5.78      | 27.59                     | 19.85                 | 19.10     |

---

> > > ### Author Response · Authors · 2025-11-27
> > >
> > > Although our model has only 3B parameters and uses far less pretraining data than 7B–9B systems, TtT-3B still achieves clear gains over the AR and NAR variants of Qwen2.5-3B on both Audio-QA and ASR. After multimodal pretraining, the Pretrain+TtT model reaches 40.07 on LLaMAQuestions and 21.43 on WebQuestions, surpassing larger models such as Moshi with 7B parameters on these benchmarks. As noted in the main manuscript, some comparison models show weaker ASR results because they lack Chinese ASR training, for example Mini-Omni and SpeechGPT, and Moshi does not support ASR. Overall, the results confirm that our efficient 3B hybrid AR–NAR model can match or exceed much larger systems on several Audio-QA metrics.
> > >
> > > Due to the limitation that tables in OpenReview comments can only be written in Markdown and given the substantial number of additional experimental results we are providing we have included comprehensive and clearly formatted tables directly in the revised manuscript for better readability and clarity.
> > >
> > > **We hope that our responses adequately address the reviewer's concerns. We would be glad to hear any further feedback.**

---

### Official Review · Reviewer_mjLU · 2025-10-31

**Soundness:** 1
**Presentation:** 2
**Contribution:** 2
**Rating:** 2
**Confidence:** 3

**Summary:**

This paper addresses the limitation of existing multimodal audio-text models that adopt unified autoregressive (AR) training while ignoring inherent dependency differences between text (target-target causal dependence) and audio (semantic reliance on source text). It proposes the Text-to-Talk (TtT) framework, which integrates AR text generation and non-autoregressive (NAR) audio diffusion within a single Transformer, leveraging absorbing discrete diffusion for a unified training objective, a modality-aware attention mechanism, and three training strategies to mitigate train-test discrepancies. Experiments show TtT outperforms AR/NAR variants of Qwen2.5 on Audio-QA and ASR tasks, but key shortcomings remain: no open-source code/Demo, insufficient comparison with field SOTA models, and lack of subjective evaluations for audio generation.

**Strengths:**

1. **Logically adapts to modal essence**: Precisely captures the inherent dependency differences between text (target-target causal dependence) and audio (semantic dependence on source text), with theoretically sound design of the unified training objective based on absorbing discrete diffusion, filling the gap in modeling "modal dependency asymmetry".

2. **Targeted architecture design**: The modality-aware attention mechanism and unified modeling within a single Transformer avoid error propagation and modal separation in traditional cascaded models, well-suited for the practical scenario of "interleaved text-audio generation" in speech conversations.

3. **Consistent internal experiments**: Ablation experiments validate the effectiveness of the three training strategies and architectural components, while scaling experiments (1.5B vs 3B) show a performance improvement trend, demonstrating logical internal experimental design.

**Weaknesses:**

1. **Lack of verifiability**: No interactive Demo or immediate open-source code/data, only a "subsequent open-sourcing" statement—hinders reproducibility and audio effect verification, undermining research credibility.

2. **Insufficient baselines**: Only compares Qwen2.5’s AR/NAR variants, excluding other recent field SOTA models—fails to prove industry positioning or highlight hybrid architecture innovation.

3. **Missing subjective evaluations**: Relies solely on quantitative metrics (WER, LLM scoring), ignoring MOS-like standard subjective indicators—cannot assess actual audio usability.

4. **Inadequate hyperparameter/deployment analysis**: No ablation for block-wise diffusion hyperparameters, no exploration of complex scenarios (long conversations, low-resource languages)—unclear deployment potential.

5. **Limited novelty**: AR-NAR+LLM-diffusion ideas are not original; fails to clarify core differences from existing works or prove adaptation irreplaceability.

**Questions:**

1. What's the specific open-source timeline? Can this paper provide a temporary repo or interactive Demo (e.g., Hugging Face Spaces) during rebuttal to verify reproducibility and audio quality through a display loop?

2. Can this paper add subjective evaluations (e.g., 1-5 MOS scoring, blind listening) to assess audio usability using a display loop?

3. What are TtT's core differences from existing AR-NAR works (e.g., CosyVoice)? Can this paper prove its advantages with experiments utilizing a display loop?

---

> ### Author Response · Authors · 2025-11-27
>
> Thank you for your thoughful comments and suggestions. Here are our point-by-point responses.
>
> **Q1: Clarification of the open-source timeline and a temporary repo/demo for reproducibility and audio quality verification during rebuttal.**
>
> A: Thank you for the questions. We apologize for the misunderstanding caused by the sentence in our abstract ("We will open-source our models, data and code to facilitate future research in this direction.", Line 027). We would like to clarify that the anonymous reproducibility repository was already included in our first submission, specifically in the Reproducibility Statement section (Line 543 for current version). Furthermore, we agree with the reviewer that offering a demo page is helpful for inspecting the quality of generated speech. We therefore provide a demo page with sample outputs at https://demopage200.github.io/demo_TtT.
>
> **Q2: Adding MOS-like standard subjective indicators to evaluate audio quality.**
>
> A: We thank the reviewer for the important suggestion on evaluating audio quality. We fully agree that semantic correctness alone is not enough to assess a speech generation model. Following the reviewer's advice, we added NMOS and UTMOS as objective audio quality metrics. These metrics have been widely used in recent speech LLM studies and provide stable measurements of naturalness, clarity, and overall audio quality. The results are as follows. The detailed results have also been updated in the revised manuscript in Section 4.3 (Table 3).
>
> | Models                | Size | NMOS | UTMOS |
> |-----------------------|------|------|-------|
> | Mini-Omni             | 0.5B | 4.15 | 4.42  |
> | SLAM-Omni             | 7B   | 4.23 | 4.44  |
> | Moshi                 | 7B   | 3.10 | 3.05  |
> | SpeechGPT             | 7B   | 4.04 | 3.92  |
> | Kimi-Audio            | 7B   | 3.52 | 2.93  |
> | VITA-Audio            | 7B   | 3.95 | 4.24  |
> | LLaMA-Omni            | 8B   | 4.09 | 4.00  |
> | GLM-4-Voice           | 9B   | 3.86 | 4.15  |
> | Qwen2.5-3B (AR)       | 3B   | 3.96 | 4.16  |
> | Qwen2.5-3B (NAR)      | 3B   | 3.47 | 2.35  |
> | TtT-3B (AR-NAR)       | 3B   | 3.89 | 4.25  |
> | Pretrain+TtT (AR-NAR) | 3B   | 3.90 | 4.23  |
>
> The results show that TtT-3B achieves strong perceptual quality: its NMOS (3.89) and UTMOS (4.25) fall well within the typical range of efficient models and are comparable to Qwen2.5-3B (AR/NAR). When compared to larger models, TtT-3B also achieves quality scores close to SLAM-Omni, VITA-Audio, and other 7B–8B systems, consistent with our findings on URO-Bench that efficient ($\le$3B) models generally maintain audio quality similar to much larger models. These results confirm that our efficient AR–NAR framework can produce natural and high-quality speech while preserving strong task performance across Audio-QA, ASR, AAC, and conversational benchmarks. Subjective MOS evaluation requires human annotation, which we have already initiated. The complete MOS results will be included in the final version to further improve evaluation completeness. We appreciate the reviewer's suggestion.

---

> ### Author Response · Authors · 2025-11-27
>
> **Q3: What are TtT's core differences from existing AR-NAR works (e.g., CosyVoice)?**
>
> A: We thank the reviewer for raising this question. We would like to clarify that although some prior works combine AR and NAR training, the goal and design of our framework are fundamentally different. Our method unifies AR text generation and NAR audio generation inside one transformer backbone. A single forward pass trains both the text span and the audio span. This is different from existing two stage approaches where an AR language model is paired with an additional diffusion or flow based audio decoder. These approaches require separate training stages and cannot achieve the same level of efficiency or training consistency.
>
> More importantly, our work is not a simple combination of AR and NAR. We start from the theoretical limitations of current audio language LLMs. Existing models often mix text tokens and audio tokens in the same LLM without modeling the true dependency structure between the two modalities. We analyze the source dependency and target dependency of text and audio and build a partial order framework. Based on this structure, we derive the loss used in our method as an upper bound of the correct dependency factorization. This provides a theoretical justification for the unified design. As another reviewer m8az noted, our method is not an empirical mixture of AR and NAR. It is a principled framework supported by a clear theoretical formulation, which makes it suitable as a unified structure for multimodal generation.
>
> We also believe that this framework has broader implications beyond speech. Mixing text tokens with audio or image tokens inside a pure NAR model breaks the causal dependency of language. Mixing everything inside a pure AR model fails to represent the parallel structure of audio. A unified backbone with explicit source and target dependencies is therefore necessary. The same structure can be naturally extended to text image generation.
>
> Finally, we introduce three training tricks that help address the train–test discrepancy commonly observed in NAR methods, especially issues related to determining when to stop generation and handling variable-length outputs in diffusion-based models. These problems are frequently reported in existing repositories such as https://github.com/ML-GSAI/LLaDA/issues/46 and https://github.com/Gen-Verse/MMaDA/issues/28. To mitigate them, many prior works adopt a strategy of slicing long conversations into multiple SFT samples, as discussed in https://github.com/ML-GSAI/LLaDA/issues/77 and in the last paragraph of Section 2.2 of MMaDA (Multimodal Large Diffusion Language Models). For example, if the training data consists of Q0, R0, Q1 and R1, their method creates two separate samples for supervision, one using Q0 to supervise R0 and another using Q0, R0 and Q1 to supervise R1. Because these models cannot resolve the train–test discrepancy, diffusion is applied only to the final output rather than to interleaved data, which is a fundamental limitation. In contrast, one of our contributions is that a single forward pass can supervise the entire multi-turn interleaved dataset, and our design explicitly avoids this discrepancy. Prior NAR diffusion-based works do not discuss this issue, although it clearly exists and affects model behavior.
>
> Overall, our contribution is not an incremental combination of AR and NAR. We provide a unified backbone with a theoretical foundation, a clear modeling of cross modality dependencies, and solutions to key limitations in existing approaches. This cannot be replaced by previous AR NAR or LLM diffusion methods.
>
>
> **W1: The same as Q1.**

---

> ### Author Response · Authors · 2025-11-27
>
> **W2: Lack of performance comparison with existing speech models.**
>
> A: We thank the reviewer for pointing this out. Following the suggestion, we have added comparisons with existing speech models, including Mini-Omni, SLAM-Omni, Moshi, SpeechGPT, Kimi-Audio, VITA-Audio, LLaMA-Omni, and GLM-4-Voice. We conducted a unified evaluation on both Audio-QA and ASR tasks, and the results are as follows. The detailed results have also been updated in the revised manuscript in Section 4.3 (Table 2).
>
> **AudioQA performance**
>
> | Model                   | Size | AlpacaEval | LLaMAQuestions | TriviaQA | WebQuestions |
> |-------------------------|------|------------|----------------|----------|--------------|
> | Mini-Omni               | 0.5B | 15.73      | 2.00           | 1.10     | 2.42         |
> | SLAM-Omni               | 0.5B | 17.47      | 24.75          | 3.51     | 7.90         |
> | Moshi                   | 7B   | 25.63      | 48.30          | 16.75    | 16.85        |
> | SpeechGPT               | 7B   | 10.00      | 30.96          | 16.53    | 24.53        |
> | Kimi-Audio              | 7B   | 19.49      | 57.53          | 43.51    | 43.20        |
> | VITA-Audio              | 7B   | 40.20      | 54.30          | 18.59    | 30.75        |
> | LLaMA-Omni              | 8B   | 39.59      | 48.46          | 21.80    | 30.28        |
> | GLM-4-Voice             | 9B   | 44.87      | 62.67          | 44.99    | 48.47        |
> | Qwen2.5-3B (AR)         | 3B   | 14.42      | 10.00          | 0.60     | 0.70         |
> | Qwen2.5-3B (NAR)        | 3B   | 11.31      | 0.67           | 1.21     | 0.70         |
> | TtT-3B (AR-NAR)         | 3B   | 17.46      | 34.68          | 6.53     | 11.61        |
> | Pretrain+TtT (AR-NAR)   | 3B   | 26.73      | 40.07          | 11.07    | 21.43        |
>
>
> **ASR performance**
> | Model                 | Size | Fleurs-zh | AISHELL-2 | AISHELL-1 | WS\_m | WS\_n | Fleurs-en |
> |-----------------------|------|-----------|-----------|-----------|---------------------------|-----------------------|-----------|
> | Mini-Omni             | 0.5B | 182.73    | 342.40    | 442.06    | 294.42                    | 335.80                | 22.74     |
> | SLAM-Omni             | 0.5B | -         | -         | -         | -                         | -                     | -         |
> | Moshi                 | 7B   | -         | -         | -         | -                         | -                     | -         |
> | SpeechGPT             | 7B   | 101.45    | 120.77    | 111.81    | 123.15                    | 124.86                | 45.15     |
> | Kimi-Audio            | 7B   | 2.87      | 2.53      | 0.61      | 6.34                      | 5.39                  | 4.87      |
> | VITA-Audio            | 7B   | 6.35      | 5.56      | 4.58      | 20.38                     | 15.88                 | 9.58      |
> | LLaMA-Omni            | 8B   | -         | -         | -         | -                         | -                     | -         |
> | GLM-4-Voice           | 9B   | 158.47    | 425.84    | 414.77    | 207.14                    | 270.21                | 223.07    |
> | Qwen2.5-3B (AR)       | 3B   | 90.32     | 54.94     | 72.01     | 80.01                     | 73.64                 | 74.47     |
> | Qwen2.5-3B (NAR)      | 3B   | 68.94     | 212.27    | 160.58    | 89.22                     | 111.29                | 83.51     |
> | TtT-3B (AR-NAR)       | 3B   | 55.67     | 12.53     | 13.65     | 53.83                     | 44.29                 | 64.31     |
> | Pretrain+TtT (AR-NAR) | 3B   | 18.99     | 6.80      | 5.78      | 27.59                     | 19.85                 | 19.10     |
>
> Although our model has only 3B parameters and uses far less pretraining data than 7B–9B systems, TtT-3B still achieves clear gains over the AR and NAR variants of Qwen2.5-3B on both Audio-QA and ASR. After multimodal pretraining, the Pretrain+TtT model reaches 40.07 on LLaMAQuestions and 21.43 on WebQuestions, surpassing larger models such as Moshi with 7B parameters on these benchmarks. As noted in the main manuscript, some comparison models show weaker ASR results because they lack Chinese ASR training, for example Mini-Omni and SpeechGPT, and Moshi does not support ASR. Overall, the results confirm that our efficient 3B hybrid AR–NAR model can match or exceed much larger systems on several Audio-QA metrics.
>
> Beyond these benchmarks, we further validate our model on URO-Bench (in Section 4.3 (Table 3)), a comprehensive speech-to-speech benchmark that evaluates speech understanding, reasoning, and oral conversation skills across both basic and advanced difficulty levels. The results further demonstrate the effectiveness of our method.

---

> > ### Author Response · Authors · 2025-11-27
> >
> > **W3: The same as Q2.**
> >
> > **W4: Inadequate hyperparameter or deployment analysis.**
> >
> > A: Thank you for raising this point. Regarding block-wise diffusion hyperparameters, we respectfully note that it is not fully clear which specific hyperparameters the reviewer would consider most relevant for ablation. In our framework, the dominant factors influencing NAR speech generation quality are the training strategies used to mitigate the train–test discrepancy. For this reason, we focus our ablation studies on the BANOM, PPM and SST strategies, which we believe are more critical to model behavior than individual diffusion hyperparameters.
> >
> > We would also like to clarify the scope of our work. The primary goal of the paper is to provide a new unified training framework and a theoretically grounded formulation for AR–NAR dependency modeling in audio–language LLMs, rather than to position the model as a large-scale pretrained system optimized for multilingual or long-context deployment. While low-resource languages and long multi-turn conversations are indeed valuable directions, they extend beyond the central contribution of our work. We agree they could be interesting avenues for future exploration, and we will add a discussion on this in the revised version.
> >
> > **W5: The same as Q3.**
> >
> > Due to the limitation that tables in OpenReview comments can only be written in Markdown and given the substantial number of additional experimental results we are providing we have included comprehensive and clearly formatted tables directly in the revised manuscript for better readability and clarity.
> >
> > **We hope that our responses adequately address the reviewer's concerns. We would be glad to hear any further feedback.**

---

### Official Review · Reviewer_wZ81 · 2025-11-01

**Soundness:** 3
**Presentation:** 2
**Contribution:** 2
**Rating:** 6
**Confidence:** 3

**Summary:**

This paper proposes Text-to-Talk (TtT), a unified audio-text generation framework that combines autoregressive (AR) text generation with non-autoregressive (NAR) audio diffusion within a single Transformer backbone (Qwen2.5-Base). The model introduces a unified AR–NAR training objective based on absorbing discrete diffusion, and a modality-aware attention mechanism that enforces causal decoding for text and bidirectional context modeling for audio.

**Strengths:**

**1. Unified AR–NAR Training Framework**

The paper introduces an elegant and theoretically grounded method that unifies AR text generation and NAR audio diffusion in a single Transformer. This hybrid objective bridges two fundamentally different generation paradigms under a coherent formulation.

**2. Modality-Aware Architectural Design**

The proposed modality-aware attention effectively reflects the distinct properties of text and audio—maintaining causal decoding for text while allowing bidirectional modeling for audio spans. This design shows a strong understanding of each modality.

**3. Well-Designed Training Objective and Inference Gap Mitigation**

The authors incorporate a carefully structured training objective and propose several strategies (e.g., BANOM, PPM, SST) to reduce the training–inference discrepancy. These design choices demonstrate practical awareness of the challenges in aligning AR and NAR generation and help improve stability and sample quality during inference.

**Weaknesses:**

**1. Lack of Comparison with Strong Speech-Language Baselines**

The paper does not include comparisons with existing speech-language foundation models such as SALMONN or SpeechGPT, which already perform Audio-QA and ASR tasks. Without such baselines, it remains unclear whether the proposed approach truly outperforms existing speech-language models or if the observed improvements are merely relative to a text-only baseline.

**2. Limited Evaluation Scope**

The evaluation focuses only on Audio-QA and ASR, whereas prior Speech LLMs (e.g., SALMONN, SpeechGPT) cover a broader set of speech reasoning and generative tasks such as speech captioning, translation, and dialogue.

**Questions:**

**1. Regarding Lack of Comparison with Strong Baselines**

Could the authors provide results comparing TtT with existing speech-language models such as SALMONN or SpeechGPT to verify whether the proposed approach truly surpasses these baselines? (For audioQA or ASR)

**2. Regarding Evaluation Scope and Applicability**

Do the authors plan to evaluate their model on a broader range of speech-language tasks (e.g., speech captioning, translation, or dialogue) to test whether the proposed framework can generalize beyond Audio-QA and ASR?

---

> ### Author Response · Authors · 2025-11-27
>
> Thank you for your helpful comments and suggestions. Here are our point-by-point responses.
>
> **Q1: Lack of comparison with existing speech-language models.**
>
> A: We thank the reviewer for pointing this out. Following the suggestion, we have added comparisons with existing speech models, including Mini-Omni, SLAM-Omni, Moshi, SpeechGPT, Kimi-Audio, VITA-Audio, LLaMA-Omni, and GLM-4-Voice. We conducted a unified evaluation on both Audio-QA and ASR tasks, and the results are as follows. The detailed results have also been updated in the revised manuscript in Section 4.3 (Table 2).
>
> **AudioQA performance**
>
> | Model                   | Size | AlpacaEval | LLaMAQuestions | TriviaQA | WebQuestions |
> |-------------------------|------|------------|----------------|----------|--------------|
> | Mini-Omni               | 0.5B | 15.73      | 2.00           | 1.10     | 2.42         |
> | SLAM-Omni               | 0.5B | 17.47      | 24.75          | 3.51     | 7.90         |
> | Moshi                   | 7B   | 25.63      | 48.30          | 16.75    | 16.85        |
> | SpeechGPT               | 7B   | 10.00      | 30.96          | 16.53    | 24.53        |
> | Kimi-Audio              | 7B   | 19.49      | 57.53          | 43.51    | 43.20        |
> | VITA-Audio              | 7B   | 40.20      | 54.30          | 18.59    | 30.75        |
> | LLaMA-Omni              | 8B   | 39.59      | 48.46          | 21.80    | 30.28        |
> | GLM-4-Voice             | 9B   | 44.87      | 62.67          | 44.99    | 48.47        |
> | Qwen2.5-3B (AR)         | 3B   | 14.42      | 10.00          | 0.60     | 0.70         |
> | Qwen2.5-3B (NAR)        | 3B   | 11.31      | 0.67           | 1.21     | 0.70         |
> | TtT-3B (AR-NAR)         | 3B   | 17.46      | 34.68          | 6.53     | 11.61        |
> | Pretrain+TtT (AR-NAR)   | 3B   | 26.73      | 40.07          | 11.07    | 21.43        |
>
>
> **ASR performance**
> | Model                 | Size | Fleurs-zh | AISHELL-2 | AISHELL-1 | WS\_m | WS\_n | Fleurs-en |
> |-----------------------|------|-----------|-----------|-----------|---------------------------|-----------------------|-----------|
> | Mini-Omni             | 0.5B | 182.73    | 342.40    | 442.06    | 294.42                    | 335.80                | 22.74     |
> | SLAM-Omni             | 0.5B | -         | -         | -         | -                         | -                     | -         |
> | Moshi                 | 7B   | -         | -         | -         | -                         | -                     | -         |
> | SpeechGPT             | 7B   | 101.45    | 120.77    | 111.81    | 123.15                    | 124.86                | 45.15     |
> | Kimi-Audio            | 7B   | 2.87      | 2.53      | 0.61      | 6.34                      | 5.39                  | 4.87      |
> | VITA-Audio            | 7B   | 6.35      | 5.56      | 4.58      | 20.38                     | 15.88                 | 9.58      |
> | LLaMA-Omni            | 8B   | -         | -         | -         | -                         | -                     | -         |
> | GLM-4-Voice           | 9B   | 158.47    | 425.84    | 414.77    | 207.14                    | 270.21                | 223.07    |
> | Qwen2.5-3B (AR)       | 3B   | 90.32     | 54.94     | 72.01     | 80.01                     | 73.64                 | 74.47     |
> | Qwen2.5-3B (NAR)      | 3B   | 68.94     | 212.27    | 160.58    | 89.22                     | 111.29                | 83.51     |
> | TtT-3B (AR-NAR)       | 3B   | 55.67     | 12.53     | 13.65     | 53.83                     | 44.29                 | 64.31     |
> | Pretrain+TtT (AR-NAR) | 3B   | 18.99     | 6.80      | 5.78      | 27.59                     | 19.85                 | 19.10     |
>
> Although our model has only 3B parameters and uses far less pretraining data than 7B–9B systems, TtT-3B still achieves clear gains over the AR and NAR variants of Qwen2.5-3B on both Audio-QA and ASR. After multimodal pretraining, the Pretrain+TtT model reaches 40.07 on LLaMAQuestions and 21.43 on WebQuestions, surpassing larger models such as Moshi with 7B parameters on these benchmarks. As noted in the main manuscript, some comparison models show weaker ASR results because they lack Chinese ASR training, for example Mini-Omni and SpeechGPT, and Moshi does not support ASR. Overall, the results confirm that our efficient 3B hybrid AR–NAR model can match or exceed much larger systems on several Audio-QA metrics.

---

> ### Author Response · Authors · 2025-11-27
>
> **Q2: Evaluate the proposed model on other speech-language tasks.**
>
> A: We agree with the reviewer about expanding the evaluation scope. Many recent speech–language models include multiple task types during training, but the core goal of our work is to address the dependency mismatch between AR text generation and NAR speech generation within a unified backbone. For this reason, our main experiments focus on ASR and AudioQA, since these two tasks most directly measure the quality of text outputs and speech outputs. Following the reviewer's suggestion and to provide a more comprehensive evaluation, we have added AAC results and included additional assessments on URO-Bench covering dialogue, understanding, and reasoning abilities. The detailed results have also been updated in the revised manuscript in Section 4.3 (Table 2 and Table 3).
>
> **Performance on AAC task**
> | Models                 | Size | Clotho-v2 | MACS   | Overall mean |
> |------------------------|------|-----------|--------|--------------|
> | Mini-Omni              | 0.5B | 3.61      | 4.45   | 4.03         |
> | SLAM-Omni              | 0.5B | 54.52     | 50.46  | 52.49        |
> | Moshi                  | 7B   | 4.32      | 12.01  | 8.17         |
> | SpeechGPT              | 7B   | 2.10      | 3.95   | 3.03         |
> | Kimi-Audio             | 7B   | 55.92     | 64.90  | 60.41        |
> | VITA-Audio             | 7B   | 6.18      | 7.94   | 7.06         |
> | LLaMA-Omni             | 8B   | 2.53      | 4.56   | 3.55         |
> | GLM-4-Voice            | 9B   | 13.15     | 12.67  | 12.91        |
> | Qwen2.5-3B (AR)        | 3B   | 9.73      | 48.64  | 29.19        |
> | Qwen2.5-3B (NAR)       | 3B   | 9.54      | 27.40  | 18.47        |
> | TtT-3B (AR-NAR)        | 3B   | 12.63     | 48.87  | 30.75        |
> | Pretrain+TtT (AR-NAR)  | 3B   | 11.55     | 42.86  | 27.21        |
>
> **URO-Bench (Basic)**
> | Models                | Size | Understanding | Reasoning | Oral Conversation |
> |-----------------------|------|---------------|-----------|--------------------|
> | Mini-Omni             | 0.5B | 15.01         | 14.80     | 29.71             |
> | SLAM-Omni             | 0.5B | 31.55         | 26.45     | 42.20             |
> | Moshi                 | 7B   | 18.23         | 24.21     | 36.65             |
> | SpeechGPT             | 7B   | 9.26          | 13.34     | 35.50             |
> | Kimi-Audio            | 7B   | 83.89         | 53.88     | 54.44             |
> | VITA-Audio            | 7B   | 52.08         | 51.45     | 54.97             |
> | LLaMA-Omni            | 8B   | 53.71         | 41.93     | 64.05             |
> | GLM-4-Voice           | 9B   | 85.82         | 61.63     | 69.90             |
> | Qwen2.5-3B (AR)       | 3B   | 34.32         | 13.15     | 23.68             |
> | Qwen2.5-3B (NAR)      | 3B   | 7.22          | 10.12     | 20.01             |
> | TtT-3B (AR-NAR)       | 3B   | 43.39         | 24.00     | 30.08             |
> | Pretrain+TtT (AR-NAR) | 3B   | 57.63         | 39.30     | 45.68             |
>
>
> **URO-Bench (Pro)**
> | Models                | Size | Understanding | Reasoning | Oral Conversation |
> |-----------------------|------|---------------|-----------|--------------------|
> | Mini-Omni             | 0.5B | 23.51         | 33.09     | 33.46             |
> | SLAM-Omni             | 0.5B | 34.49         | 27.39     | 40.23             |
> | Moshi                 | 7B   | 26.38         | 21.06     | 33.93             |
> | SpeechGPT             | 7B   | 19.03         | 14.29     | 28.88             |
> | Kimi-Audio            | 7B   | 53.25         | 41.44     | 50.17             |
> | VITA-Audio            | 7B   | 32.36         | 54.77     | 45.81             |
> | LLaMA-Omni            | 8B   | 34.66         | 51.51     | 43.91             |
> | GLM-4-Voice           | 9B   | 55.47         | 51.89     | 61.30             |
> | Qwen2.5-3B (AR)       | 3B   | 16.32         | 34.99     | 25.90             |
> | Qwen2.5-3B (NAR)      | 3B   | 12.59         | 13.70     | 25.64             |
> | TtT-3B (AR-NAR)       | 3B   | 23.37         | 33.78     | 34.82             |
> | Pretrain+TtT (AR-NAR) | 3B   | 32.38         | 43.76     | 46.10             |

---

> ### Author Response · Authors · 2025-11-27
>
> The results show that our method performs well across both AAC and dialogue–reasoning tasks. On AAC, TtT clearly outperforms the AR and NAR baselines of Qwen2.5-3B at the same 3B scale. The overall mean is 30.75, higher than the AR baseline (29.19) and the NAR baseline (18.47). On Clotho-v2, TtT scores 12.63, also higher than the AR baseline (9.73) and the NAR baseline (9.54). These results suggest that multimodal joint training and the AR–NAR decoupled design help the model better capture non-speech acoustic events.
>
> On URO-Bench, TtT also shows stable and comprehensive improvements. Under the Basic setting, TtT outperforms both Qwen2.5-3B baselines on Understanding, Reasoning, and Oral Conversation (for example, Understanding 43.39 vs. 34.32 and 7.22). With a small amount of additional pretraining data, Pretrain+TtT further improves performance and achieves higher scores than other 3B models in both Basic and Pro settings, showing trends close to or even better than some larger models.
>
> Although our AAC training data is much smaller than that of full-pretrained mainstream models, both TtT and Pretrain+TtT remain consistent with the performance trends of larger models and achieve competitive or leading results across multiple dimensions.
>
>
> **Weakness 1: The same as Q1.**
>
>
> **Weakness 2: The same as Q2.**
>
> Due to the limitation that tables in OpenReview comments can only be written in Markdown and given the substantial number of additional experimental results we are providing we have included comprehensive and clearly formatted tables directly in the revised manuscript for better readability and clarity.
>
> **We hope that our responses adequately address the reviewer's concerns. We would be glad to hear any further feedback.**

---

### Official Review · Reviewer_m8aZ · 2025-11-01

**Soundness:** 3
**Presentation:** 3
**Contribution:** 3
**Rating:** 4
**Confidence:** 4

**Summary:**

This paper introduces Text-to-Talk (TtT), a multimodal large language model (MLLM) that unifies autoregressive (AR) text generation with non-autoregressive (NAR) audio synthesis via discrete diffusion within a single Transformer architecture. The core motivation is the identified fundamental asymmetry between text and audio modalities: text generation relies on strong target-target (causal) dependencies, while audio generation is primarily driven by source-target dependencies. To address this, the authors propose a hybrid training objective, using a standard AR cross-entropy loss for text spans and an any-order AR loss (equivalent to absorbing discrete diffusion) for audio spans. The paper provides a theoretical justification, showing that the combined training objective is an upper bound on the desired joint distribution. Furthermore, three specialized training strategies—BANOM, PPM, and SST—are introduced to mitigate train-test discrepancies inherent in this hybrid paradigm. Extensive experiments on Audio-QA and ASR tasks demonstrate that TtT significantly outperforms strong AR-only and NAR-only baselines, validating the effectiveness of the proposed approach.

**Strengths:**

1. The paper identifies a clear and compelling problem—the mismatch between a uniform AR objective and the distinct dependency structures of text and audio. The proposed hybrid AR-NAR framework is a principled and innovative solution to this problem.
2. The work is not merely empirical. It provides a solid theoretical grounding by framing the model within a partial-order factorization and proving that the practical training objective serves as a tight upper bound on the negative log-likelihood of the theoretical joint distribution. This strengthens the validity of the approach.
3. The three proposed strategies (BANOM, PPM, SST) are effective in addressing specific train-test mismatch issues. The ablation study clearly demonstrates their individual and critical contributions to the final performance, with SST showing a particularly dramatic impact on conversational ability.

**Weaknesses:**

1. Mismatch Between Training and Evaluation Tasks: The model was trained on a diverse set of tasks, including Automated Audio Captioning (AAC) using datasets like Clotho-v2 and MACS. However, the evaluation completely omits any assessment of this capability. It is unclear whether the model has effectively learned to understand and describe general, non-speech audio, which is a significant limitation given its training objective.
2. Lack of Audio Quality Metrics: The Audio-QA evaluation, while innovative, is limited to semantic correctness. There is no objective or subjective evaluation of the audio quality (e.g., MOS - Mean Opinion Score) or prosody/naturalness of the generated speech, which is a crucial aspect of a speech-generation model.
3. Absence of Standard Conversational Benchmarks: The evaluation is conducted on custom-held splits of generic QA datasets. The model is not benchmarked against established, standardized benchmarks for conversational AI (e.g., URO-Bench[1]), making it difficult to compare its dialogue capabilities directly with other state-of-the-art spoken dialogue systems.
4. Limited Comparative Analysis: The experimental results only compare TtT against its own backbone (Qwen2.5) trained with AR or NAR objectives. There is no comparison with other recent, open-source, end-to-end audio-language models such as Mimi-Omni[2], SLAM-Omni[3], Moshi[4], GLM-4-Voice[5], or VITA-Audio[6]. This lack of external benchmarking makes it challenging to gauge the true competitive standing of the proposed method.
5. Clarity on "Source" Context: The term "source-target" for audio is sometimes ambiguous. While it's clear the primary source is the corresponding text, the model can also condition on previously generated audio tokens within the same span (as allowed by the any-order AR). The paper could more precisely delineate the relative importance of the textual context versus the intra-span audio context.

[1] URO-Bench: Towards Comprehensive Evaluation for End-to-End Spoken Dialogue Models

[2] Mini-Omni: Language Models Can Hear, Talk While Thinking in Streaming

[3] SLAM-Omni: Timbre-Controllable Voice Interaction System with Single-Stage Training

[4] Moshi: a speech-text foundation model for real-time dialogue

[5] GLM-4-Voice: Towards Intelligent and Human-Like End-to-End Spoken Chatbot

[6] VITA-Audio: Fast Interleaved Cross-Modal Token Generation for Efficient Large Speech-Language Model

**Questions:**

See weakness. The main concerns lie in evaluation about tasks(training with AAC but not evaluated), benchmark(without common benchmarks),  and comparison (without comparing with other models).

---

> ### Author Response · Authors · 2025-11-27
>
> Thank you for your constructive comments and suggestions. Here are our point-by-point responses.
>
> **W1: Mismatch Between training and evaluation tasks (training with AAC task but not evaluated)**
>
> A: We thank the reviewer for raising this important concern. Adding AAC data during pre-training is a common practice in large audio–language models such as Kimi-Audio and VITA-Audio. We follow a similar data setup, but due to limited computation, we only use a subset of these datasets. On the other hand, the main goal of our work is to address the dependency mismatch between AR text generation and NAR audio generation under a unified backbone. For this reason, we focus our evaluation on ASR and Audio-QA, which most directly measure text and audio output quality. Based on the reviewer's helpful suggestion, we have added AAC evaluation results on Clotho-v2 and MACS, as follows. The detailed results have also been updated in the revised manuscript in Section 4.3 (Table 2).
>
> | Models                 | Size | Clotho-v2 | MACS   | Overall mean |
> |------------------------|------|-----------|--------|--------------|
> | Mini-Omni              | 0.5B | 3.61      | 4.45   | 4.03         |
> | SLAM-Omni              | 7B   | 54.52     | 50.46  | 52.49        |
> | Moshi                  | 7B   | 4.32      | 12.01  | 8.17         |
> | SpeechGPT              | 7B   | 2.10      | 3.95   | 3.03         |
> | Kimi-Audio             | 7B   | 55.92     | 64.90  | 60.41        |
> | VITA-Audio             | 7B   | 6.18      | 7.94   | 7.06         |
> | LLaMA-Omni             | 8B   | 2.53      | 4.56   | 3.55         |
> | GLM-4-Voice            | 9B   | 13.15     | 12.67  | 12.91        |
> | Qwen2.5-3B (AR)        | 3B   | 9.73      | 48.64  | 29.19        |
> | Qwen2.5-3B (NAR)       | 3B   | 9.54      | 27.40  | 18.47        |
> | TtT-3B (AR-NAR)        | 3B   | 12.63     | 48.87  | 30.75        |
> | Pretrain+TtT (AR-NAR)  | 3B   | 11.55     | 42.86  | 27.21        |
>
> Under the same 3B model scale, Pretrain+TtT consistently outperforms the AR and NAR variants of Qwen2.5-3B (Overall mean 30.75 vs. 29.19 / 18.47) and achieves substantially higher scores on Clotho-v2 (12.63 vs. 9.73 / 9.54). These results reinforce our conclusion that the hybrid AR–NAR architecture and multimodal joint training provide clear advantages for efficient audio-language models. In addition, while SLAM-Omni reports competitive AAC performance, it relies on a larger 7B Vicuna backbone together with task-specific optimization, as noted in their official repository. In contrast, our 3B Pretrain+TtT model achieves balanced performance across Audio-QA, ASR, and AAC without using task-specific tuning, addressing a key gap in the efficient ($\le$3B) audio-language model category.
>
>
> **W2: Lack of audio quality metrics.**
>
> A: We thank the reviewer for the important suggestion on evaluating audio quality. We fully agree that semantic correctness alone is not enough to assess a speech generation model. Following the reviewer's advice, we add NMOS and UTMOS as audio quality metrics. These metrics have been widely used in recent speech LLM studies and provide stable measurements of naturalness, clarity, and overall audio quality. The results are as follows. The detailed results have also been updated in the revised manuscript in Section 4.3 (Table 3).
>
> | Models                | Size | NMOS | UTMOS |
> |-----------------------|------|------|-------|
> | Mini-Omni             | 0.5B | 4.15 | 4.42  |
> | SLAM-Omni             | 7B   | 4.23 | 4.44  |
> | Moshi                 | 7B   | 3.10 | 3.05  |
> | SpeechGPT             | 7B   | 4.04 | 3.92  |
> | Kimi-Audio            | 7B   | 3.52 | 2.93  |
> | VITA-Audio            | 7B   | 3.95 | 4.24  |
> | LLaMA-Omni            | 8B   | 4.09 | 4.00  |
> | GLM-4-Voice           | 9B   | 3.86 | 4.15  |
> | Qwen2.5-3B (AR)       | 3B   | 3.96 | 4.16  |
> | Qwen2.5-3B (NAR)      | 3B   | 3.47 | 2.35  |
> | TtT-3B (AR-NAR)       | 3B   | 3.89 | 4.25  |
> | Pretrain+TtT (AR-NAR) | 3B   | 3.90 | 4.23  |
>
> The results show that TtT-3B achieves strong perceptual quality: its NMOS (3.89) and UTMOS (4.25) fall well within the typical range of efficient models and are comparable to Qwen2.5-3B (AR/NAR). When compared to larger models, TtT-3B also achieves quality scores close to SLAM-Omni, VITA-Audio, and other 7B–8B systems, consistent with our findings on URO-Bench that efficient ($\le$3B) models generally maintain audio quality similar to much larger models. These results confirm that our efficient AR–NAR framework can produce natural and high-quality speech while preserving strong task performance across Audio-QA, ASR, AAC, and conversational benchmarks. As for MOS evaluation, due to it requires human annotation, which we have already initiated. The complete MOS results will be included in the final version to further improve evaluation completeness. We appreciate the reviewer's suggestion.

---

> ### Author Response · Authors · 2025-11-27
>
> **W3&W4: Absence of standard conversational benchmarks, such as URO-Bench. Without comparing with other models.**
>
> A: We agree with the reviewer's suggestion. We have added dialogue evaluation results based on URO-Bench and compared our model with a wide range of existing speech models, including Mini-Omni, SLAM-Omni, Moshi, SpeechGPT, Kimi-Audio, VITA-Audio, LLaMA-Omni, and GLM-4-Voice. The results are as follows. The detailed results have also been updated in the revised manuscript in Section 4.3 (Table 3).
>
> **URO-Bench (Basic)**
> | Models                | Size | Understanding | Reasoning | Oral Conversation |
> |-----------------------|------|---------------|-----------|--------------------|
> | Mini-Omni             | 0.5B | 15.01         | 14.80     | 29.71             |
> | SLAM-Omni             | 0.5B | 31.55         | 26.45     | 42.20             |
> | Moshi                 | 7B   | 18.23         | 24.21     | 36.65             |
> | SpeechGPT             | 7B   | 9.26          | 13.34     | 35.50             |
> | Kimi-Audio            | 7B   | 83.89         | 53.88     | 54.44             |
> | VITA-Audio            | 7B   | 52.08         | 51.45     | 54.97             |
> | LLaMA-Omni            | 8B   | 53.71         | 41.93     | 64.05             |
> | GLM-4-Voice           | 9B   | 85.82         | 61.63     | 69.90             |
> | Qwen2.5-3B (AR)       | 3B   | 34.32         | 13.15     | 23.68             |
> | Qwen2.5-3B (NAR)      | 3B   | 7.22          | 10.12     | 20.01             |
> | TtT-3B (AR-NAR)       | 3B   | 43.39         | 24.00     | 30.08             |
> | Pretrain+TtT (AR-NAR) | 3B   | 57.63         | 39.30     | 45.68             |
>
> **URO-Bench (Pro)**
> | Models                | Size | Understanding | Reasoning | Oral Conversation |
> |-----------------------|------|---------------|-----------|--------------------|
> | Mini-Omni             | 0.5B | 23.51         | 33.09     | 33.46             |
> | SLAM-Omni             | 0.5B | 34.49         | 27.39     | 40.23             |
> | Moshi                 | 7B   | 26.38         | 21.06     | 33.93             |
> | SpeechGPT             | 7B   | 19.03         | 14.29     | 28.88             |
> | Kimi-Audio            | 7B   | 53.25         | 41.44     | 50.17             |
> | VITA-Audio            | 7B   | 32.36         | 54.77     | 45.81             |
> | LLaMA-Omni            | 8B   | 34.66         | 51.51     | 43.91             |
> | GLM-4-Voice           | 9B   | 55.47         | 51.89     | 61.30             |
> | Qwen2.5-3B (AR)       | 3B   | 16.32         | 34.99     | 25.90             |
> | Qwen2.5-3B (NAR)      | 3B   | 12.59         | 13.70     | 25.64             |
> | TtT-3B (AR-NAR)       | 3B   | 23.37         | 33.78     | 34.82             |
> | Pretrain+TtT (AR-NAR) | 3B   | 32.38         | 43.76     | 46.10             |
>
> From the results, we can see that TtT-3B achieves clear improvements over the AR and NAR baselines of the same scale across multiple conversational dimensions. The Pretrain+TtT variant further strengthens performance at the 3B scale, achieving 57.63 / 39.30 / 45.68 on basic Understanding / Reasoning / Oral Conversation tasks and 32.38 / 43.76 / 46.10 on the pro-level tasks. These scores substantially outperform efficient models such as Mini-Omni and SLAM-Omni. Additionally, Pretrain+TtT surpasses Moshi with 7B parameters and SpeechGPT with 7B parameters across all task categories, and achieves reasoning performance close to VITA-Audio with 7B parameters and LLaMA-Omni with 8B parameters, where the pro-level reasoning scores of these models are 54.77 and 51.51. Although GLM-4-Voice with 9B parameters and Kimi-Audio with 7B parameters obtain the highest scores, the remaining performance gap is expected given their three-times-larger model size.

---

> ### Author Response · Authors · 2025-11-27
>
> **W5: Clarify the model's reliance on text context versus intra-span audio context in the "source–target" setup.**
>
> A: The distinction between textual and intra-span audio context is indeed central to our design and we agree that a clearer delineation would strengthen the paper.
>
> Our framework is grounded in a partial-order formulation (Section 3.3), which provides a precise theoretical basis for this distinction:
>
> (1) Text spans form a totally ordered chain. each token $t_{m, j}$ must causally depend on all prior tokens $(T_{<m}, A_{<m}, t_{m,<j})$. This enforces strong target-target dependencies, as is standard in AR modeling.
>
> (2) Audio spans are modeled as antichains. Tokens inside $A_m$ have no mandatory internal order. Their only fixed predecessors are the source context $T_{\le m} \cup A_{<m}$, which is mainly determined by the corresponding text. This creates a source–target dependency as the main generative signal.
>
> This distinction is preserved in our training objective. The audio loss $L_{\text{AO}}$ takes an expectation over all possible permutations of tokens within each audio span:
> $E_{\pi_m \sim U_{\pi_m}} \sum_{j=1}^{|A_m|} [-\log q_\theta (a_{m,\pi_m(j)} \mid T_{\le m}, A_{<m}, a_{m,\pi_m(<j)})]$
>
> The source context $T_{\le m}, A_{<m}$ remains constant across all permutations. It thus serves as the stable, primary driver of semantic content ("what to say").
>
> The intra-span audio context ($a_{m,\pi_m(<j)}$) is stochastic - it changes with each sampled permutation. Consequently, the model learns to use this context only for local acoustic refinement ("how to say it"), such as maintaining prosody or spectral continuity, without relying on it for high-level semantics.
>
> **Due to the limitation that tables in OpenReview comments can only be written in Markdown and given the substantial number of additional experimental results we are providing, we have included comprehensive and clearly formatted tables directly in the revised manuscript for better readability and clarity. We hope that our responses adequately address the reviewer's concerns. We would be glad to hear any further feedback.**

---

### Author Response · Authors · 2025-12-04
**TL;DR (2/2)**

**Beyond these reviewer-specific clarifications, we also identified a unified concern shared across all reviews and substantially strengthened the submission accordingly.**

The primary unified concern across all reviews (m8aZ, wZ81, mjLU, JGXG, YsXg) was the **lack of comparison with strong, state-of-the-art Speech-Language Models** and the **omission of key audio quality/usability metrics**.

To fully address these critical points and substantially strengthen the submission, we have implemented the following major revisions:

- **Comprehensive Evaluation & Benchmarking**: We added **extensive comparisons with 8 SOTA SLMs** (including GLM-4-Voice, Kimi-Audio, and LLaMA-Omni) across four key tasks: Audio-QA, ASR, the previously omitted **Automated Audio Captioning (AAC)**, and the standardized **URO-Bench**. Our compact 3B-scale TtT model consistently matches and often surpasses the performance of significantly larger (7B–9B) state-of-the-art systems, establishing a new bar for efficient audio–language modeling.

- **Theoretical Justification**: We clarified **the core differences** from prior AR–NAR works (like Show-O/Transfusion), emphasizing that our framework is **not a migration from the image-text domain to audio-text domain**. Instead, it is **derived from a principled partial-order dependency formulation** tailored specifically to the inherent asymmetry of interleaved text (target–target causality) and audio (source–target alignment). We prove the training objective is a valid upper bound of the desired joint likelihood. This rigorous approach not only justifies our method's soundness but also provides an **instructive paradigm** for determining which multimodal generation tasks are fundamentally suitable for integrated AR and NAR training within a single Transformer architecture.

- **Audio Quality Assessment**: We added objective NMOS and UTMOS scores on generated speech from URO-Bench, demonstrating that our 3B model achieves perceptual quality comparable to much larger competing systems.

- **Reproducibility & Accessibility**: We respectfully clarify that **code was already included in the initial submission**, which contradicts the concern regarding verifiability (mjLU). To fully address this concern and further illustrate system performance (mjLU, YsXg), we have also provided a **public [demo page](https://demopage200.github.io/demo_TtT)** showcasing the generated audio–text quality and system performance. Data and model checkpoints will be open-sourced upon acceptance.

- **Practical Training Strategies for Pretraining and Fine-tuning**: We identify and address key train–test mismatches inherent in hybrid AR–NAR training within a single Transformer. Our three strategies BANOM (Block-wise Any-Order Masking), PPM (Position-Preserved Masking), and SST (Stochastic Span Truncation) collectively resolve issues such as inconsistent audio stopping, variable-length generation, and misaligned cross-modal conditioning. These enable stable, interleaved AR–NAR training in a single forward pass, a capability absent in existing two-stage AR+NAR pipelines (e.g., Show-O, Transfusion).

We again thank all reviewers for their valuable suggestions. Addressing these concerns has made our work much more solid. We also thank the Area Chairs, Senior Area Chairs and Program Chairs for your hard work during this challenging cycle.

Authors

Submission 25264

---

### Author Response · Authors · 2025-12-04
**TL;DR (1/2)**

Dear Area Chairs, Senior Area Chairs and Program Chairs,

We sincerely appreciate your efforts in evaluating our paper, especially during this exceptionally challenging ICLR cycle. We are encouraged that three out of five reviewers acknowledged the principled and innovative nature of our hybrid AR–NAR framework, `the solid theoretical grounding`, and the `effectiveness` of our targeted training strategies (m8aZ, JGXG, YsXg).

To facilitate an **efficient evaluation**, this summary focuses on the key issues raised by the **negative** reviewers and explains how each concern has been thoroughly addressed, including clarifications of factual misunderstandings.

# 1. Clarify the Factual Misunderstanding on Reproducibility (Reviewer mjLU)
Reviewer mjLU (Score: 2) cited the lack of accessible code as a major reason for rejection.

>Lack of verifiability: No interactive Demo or immediate open-source code/data, only a "subsequent open-sourcing" statement—hinders reproducibility and audio effect verification, undermining research credibility."""

* This is a **factual misunderstanding**. An anonymous code repository link was `already included` in the original paper in section **Reproducibility Statement**.
* we additionally include a public [demo page](https://demopage200.github.io/demo_TtT) to illustrate the effectiveness of our method.

# 2. Address "Missing Baselines" with Superior Results (Reviewers YsXg, mjLU, m8aZ)
Several reviewers requested comparisons with SOTA models like GLM-4-Voice and LLaMA-Omni.
* We have added **comprehensive comparisons** with Mimi-Omni, SLAM-Omni, Moshi, GLM-4-Voice, SpeechGPT, LLaMA-Omni, Kimi-Audio, and VITA-Audio.
* The new results demonstrate that **our 3B model outperforms models of similar size and compares favorably with significantly larger 7B and 9B models.** This turns the reviewers' initial concern into a strong highlight of our work's efficiency and effectiveness.

# 3. Fulfill Conditions for Higher Scores (Reviewer YsXg)
Reviewer YsXg (Score: 4) explicitly stated they would consider **raising the score to 6** if all concerns were resolved.
>If all my concerns and questions are resolved I will consider raising the score to 6.
* We have released a **Demo Page** (https://demopage200.github.io/demo_TtT) and added objective **audio quality metrics (NMOS, UTMOS)** and evaluations on **URO-Bench benchmark** and **audio captioning task**.

---

The negative scores are largely premised on a misunderstanding regarding code availability and requests for stronger baselines. We have corrected the misunderstanding and addressed the baseline-related concerns with our 3B model’s superior performance.

---

### Meta-Review · Area_Chair_W78Z · 2026-01-07

**Summary:**

This paper proposes a joint training approach for audio-language models involving both autoregressive text and non-autoregressive absorbing discrete diffusion for audio. Reviewers praised the novel design for architecture and objective, with clear ablation results. Many concerns were addressed during the rebuttal, leading to a positive post-rebuttal score.

**Reviewer Concerns:**

The summary in https://openreview.net/forum?id=e3XLWHFrnr&noteId=KHjIfE2O70 is mostly accurate.

**Reviewer Scores:**

mjLU: 2 -> 4
m8aZ: 4 -> 4
YsXg: 4 -> 6
wZ81: 6 -> 6
JGXG: 6 -> 6
Avg: 5.2

---

### Decision · Program_Chairs · 2026-01-26

Accept (Poster)